# LOOK&LEARN: BRIDGING PERCEPTION AND GROUNDING GAP IN VISION-LANGUAGE MODELS

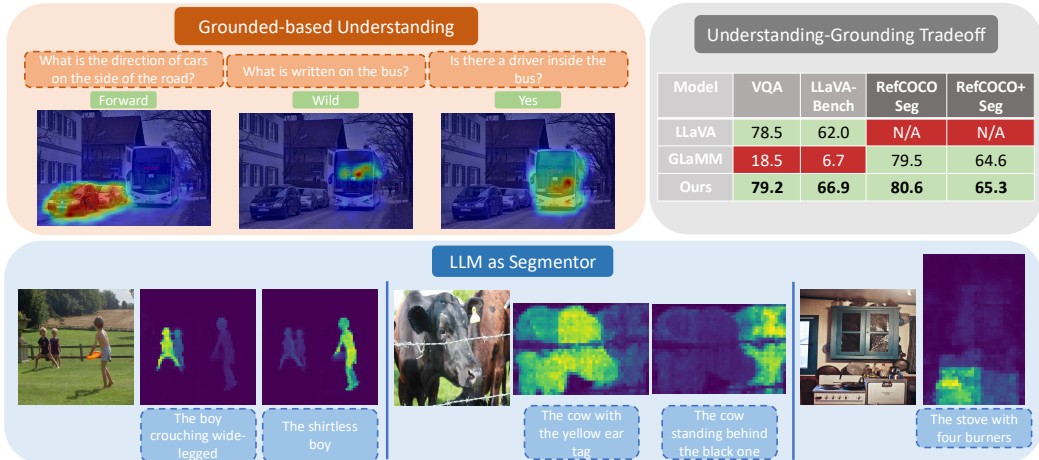

Figure 1: **Top:** Grounded-based understanding—our loss aligns attention maps with relevant visual regions during question answering, improving both grounding and perception. The accompanying table highlights the understanding–grounding tradeoff, where our method closes the gap compared to LLaVA and GLaMM. **Bottom:** LLM as Segmentor—attention maps directly serve as segmentation masks for fine-grained referring expressions, demonstrating the ability of the LLM itself to act as a segmentor without relying on heavy external decoders.

## ABSTRACT

Vision-Language Models (VLMs) excel at perception and reasoning, yet their ability to precisely ground visual concepts remains limited. A common remedy is to attach a segmentation decoder (e.g., SAM) to VLMs, which indeed equips them with segmentation capabilities but simultaneously erodes their understanding performance. We call this the *understanding–grounding gap*, where enhancing grounding comes at the expense of perception. In this work, we propose Look&Learn, a segmentation-free grounding framework that closes this gap by enabling the *LLM itself to act as a segmentor*. At the core is *Where to Look?*, a lightweight, plug-and-play loss applied directly to the attention heatmaps of arbitrary VLMs. Rather than relying on an additional decoder, our loss guides the model to focus on relevant image regions during text generation, effectively leveraging the LLM's own attention as high-quality segmentation masks. Your native LLM is your segmentor. Look&Learn is efficient and versatile: it can be integrated seamlessly into any VLM architecture and applied during pre-training, fine-tuning, or even post-training. To enable joint training of understanding and grounding, we further construct a scalable pseudo-grounding dataset that aligns textual and visual entities using state-of-the-art grounding models combined with LLM-based adjudication. Extensive experiments demonstrate that Look&Learn consistently improves grounding performance without sacrificing perception. On LLaVA, it delivers +3% gains on grounding benchmarks while also boosting understanding by 1–2%. On GLaMM, our loss further provides +1% improvements across three grounding datasets. Our gains increase as the underlying vision backbone becomes more expressive. On Qwen2.5VL-3B, the average gain rises to $\approx 5.8\%$, and on the stronger Qwen3VL-4B, equipped with a significantly more capable visual stack, the average gain further increases to $\approx 7.7\%$. These results

confirm our central hypothesis: grounding and understanding are not competing objectives but can be unified within a single model, without explicit segmentation supervision. Our work paves the way for more interpretable, efficient, and multimodal-aware VLMs, and more broadly, opens the door to using LLMs themselves as segmentors.

# 1 INTRODUCTION

**The understanding–grounding gap.** Recent advances in Vision-Language Models (VLMs) have significantly improved the ability of AI systems to perceive and understand visual content (Zhu et al., 2023; Liu et al., 2023; 2024a; Kar et al., 2024; Chen et al., 2024; Bai et al., 2025; McKinzie et al., 2024; Tong et al., 2024a; Shi et al., 2024). With their success, researchers have extended VLMs to diverse tasks such as video understanding (Yuan et al., 2025; Ataallah et al., 2024b;a;c; Maaz et al., 2023; Li et al., 2024) and multimodal generation beyond text (You et al., 2023; Tang et al., 2024; Koh et al., 2023; Ge et al., 2023; Wu et al., 2024b; Team, 2024; Tong et al., 2024b; Xie et al., 2024; Zhou et al., 2024; Dai et al., 2023; Ge et al., 2024; Wu et al., 2024a). However, integrating such additional functionalities often comes at the cost of understanding performance. This tradeoff is particularly clear when VLMs are asked to generate new modalities (e.g., masks or bounding boxes), where perception degrades in favor of new outputs. A similar challenge arises in grounding tasks, where the goal is to link generated text to specific visual regions (Zhang et al., 2024b; Xia et al., 2024; Peng et al., 2023; Lai et al., 2024; Zhang et al., 2024a; Rasheed et al., 2024; Yuan et al., 2025). Existing approaches extend VLMs with external segmentation decoders such as SAM, connected via a special [SEG] token. We argue that, this design introduces two critical issues. First, it creates a severe tradeoff: as illustrated in Figure 1, GLaMM achieves segmentation by integrating SAM with LLaVA but suffers a dramatic drop in understanding accuracy (e.g., 78.5% $\rightarrow$ 18.5% on VQA-v2). Second, such bottleneck-based integration forces the LLM to compress rich grounding signals into a single token, limiting synergy between the two tasks.

**LLM as Segmentor.** We propose the first attempt to teach the LLM itself to act as a segmentor, eliminating the need for external decoders. Instead of hurting perception, grounding and understanding can reinforce each other when combined in a shared feature space. Our core argument is that performance degradation arises not because grounding and perception are incompatible, but because existing approaches treat grounding as a separate modality generation problem. Zhang et al. (2023); Kang et al. (2025) show that the LLM's own attention maps already contain implicit grounding signals: when answering a question about an object, the model partially attends to that region. This insight motivates us to directly harness and refine these attention signals for grounding.

**Where to Look?** To realize this idea, we introduce Where-to-Look, a novel attention-guidance loss that encourages VLMs to look at the correct visual regions while generating text. For example, when answering "Yes, there is a driver in the bus," the attention map should highlight the region of the driver (Figure 1). This serves two purposes simultaneously: (1) the attention maps themselves become usable grounding masks, and (2) aligning text generation with visual focus reduces hallucinations by forcing the model to attend to relevant evidence. Our loss is lightweight, plug-and-play, and can be applied during pre-training, fine-tuning, or post-training.

**Scalable pseudo-data for grounding.** Training such a loss requires paired data linking visual words (e.g., color, position, object presence) with masks. To overcome this, we build a scalable pseudo-data pipeline that generates reliable grounding masks by combining multiple state-of-the-art grounding models with LLM-based adjudication. This enables us to train at scale without requiring ground-truth segmentation supervision.

**Consistent improvements across benchmarks.** By leveraging pseudo masks and incorporating Where-to-Look, our method bridges the understanding–grounding gap. On LLaVA Liu et al. (2024a), our approach improves understanding performance across VQA benchmarks by $\sim$1–2% while delivering +3% gains on RefCOCO Kazemzadeh et al. (2014). Compared to GLaMM, which relies on external segmentation decoders and trained on GT-masks, our model surpasses its zero-shot results on RefCOCO, RefCOCOg Mao et al. (2016), and RefCOCO+ Yu et al. (2016) by over 1%. Integrating our loss on top of GLaMM during fine-tuning or post-training enhances the performance by 1% as well. These results confirm that grounding and perception are not adversarial but mutually beneficial, and that with the right design, VLMs can unify both within a single model.

| Method | IoU | FPGS |
|--------|-----|------|
| LLaVA | 5.6 | 26.0 |
| InternVL | 7.9 | 31.7 |

Table 1: Quantitative comparison of LLaVA and InternVL's attention maps.

| Method | VQA | GQA | LLaVA-Bench | TxtQA | RefCOCO |
|--------|-----|-----|-------------|-------|---------|
| LLaVA | 78.5 | 62.0 | 63.4 | 58.0 | N/A |
| LLaVA-G | 75.1 | 58.3 | 57.9 | 54.4 | 77.1 |
| GlaMM | 18.5 | 14.6 | 6.7 | 8.4 | 79.5 |

Table 2: Comparison showing perception–grounding trade-off.

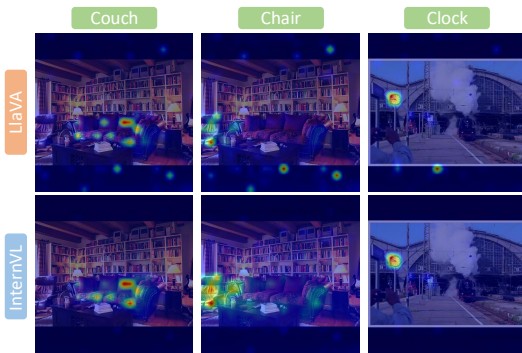

Figure 2: Qualitative grounding attention maps for LLaVA and InternVL.

**Contributions.** In summary, this work makes the following contributions: 1) We identify and address the *understanding–grounding gap* that arises when segmentation decoders are added to VLMs. 2) We introduce the concept of *LLM as Segmentor*, showing that the LLM's own attention maps can serve as segmentation masks. 3) We propose Where-to-Look, a lightweight plug-and-play loss for aligning attention with text generation. 4) We develop a scalable pseudo-data pipeline for training without ground-truth masks. 5) We demonstrate consistent improvements in both grounding and reasoning across strong baselines and diverse benchmarks.

## 2 REVISITING VLMS

This section revisits the fundamental characteristics of VLMs, highlighting their core design principles and limitations. We critically assess their ability to "see" and ground visual entities, demonstrating key weaknesses in their attention mechanisms.

### 2.1 DO VLMS SEE WELL?

One fundamental limitation of VLMs is their grounding ability—whether they attend to the correct visual regions when answering a question. To analyze this, we use LLaVA and InternVL2.5, given their widespread adoption in follow-up research. We examine their native grounding ability by extracting and analyzing the computed attention maps.

**Attention Representation in VLMs** The self-attention mechanism in an arbitrary VLM produces an attention map across all $L$ transformer layers, $Att_{Map} \in \mathbb{R}^{L \times N \times N}$, where $N$ represents the total input sequence length. To evaluate whether LLaVA attends to the correct image regions while generating answers, we select the GQA benchmark that provides VQA-style questions and GT bounding boxes. Additionally, we use the concise answers from GQA, which consist mainly of single words, allowing us to visualize attention maps for each generated answer precisely. To compute grounding attention maps, we follow these steps: 1) Aggregate attention across layers: $G_{att} = \frac{1}{L} \sum_{l=1}^{L} Att_{Map}^{(l)} \in \mathbb{R}^{N \times N}$ 2) Extract the image attention relevant to answer tokens: $G_{att} \in \mathbb{R}^{N_{ans} \times N_{img}}$, where $N_{ans}$ represents the number of tokens in the generated answer from the entire text $N_{txt}$. If $N_{ans} > 1$ (e.g., "blue car"), we average across answer tokens. 3) Compare against the Ground-Truth mask $M_{gt} \in \mathbb{R}^{H \times W}$ by upsampling $G_{att}$ to match the spatial resolution. We use IoU as a primary metric $IoU(G_{att}, M_{gt})$.

**Flexible Metric for Sparse Attention Maps** While IoU is a strong metric for dense segmentation masks, grounding attention maps are often sparse. This sparsity can cause penalties in IoU, even when the model attends to the correct regions. To address this, we propose a Flexible Precision-Aware Grounding Score (FPGS):

$$FPGS = \frac{|G_{att} \cap M_{gt}|}{|M_{gt}| + \lambda |G_{att} \setminus M_{gt}|} \tag{1}$$

, where $\lambda$ is a penalty weight controlling tolerance to false positives.

**VLM does not see.** As illustrated in Figure 2, LLaVA's native attention is diffuse and often misplaced: it attends to irrelevant areas (e.g., padded borders) and to incorrect image regions. Quantitatively, the low IoU and FPGS scores indicate weak overlap with ground-truth object masks. Although prior work Zhang et al. (2023); Kang et al. (2025) finds that certain layer–head pairs can exhibit token-relevant visual focus, reliably identifying these components is non-trivial and prompt-/image-dependent. Moreover, even when a specific layer–head combination shows partial saliency, such incidental focus does not amount to robust grounding.

## 2.2 Unsolicited Advice: Perception–Grounding Tradeoff

**Problem: adding grounding often hurts perception.** Grounding-based VLMs should ideally gain spatial reasoning without eroding core perception skills. In practice, however, we observe a consistent degradation on standard perception benchmarks once grounding is introduced.

**Setup for a fair comparison.** We compare two state-of-the-art grounding extensions—GLaMM and LLaVA-G—against the original LLaVA. These models retain LLaVA's backbone (no MoE, no stronger vision encoder), augment it with a segmentation decoder for grounding, and use the same multimodal pre-training data as LLaVA, with additional segmentation data for grounding.

**The cost of grounding.** Despite having access to *more* data, GLaMM and LLaVA-G exhibit notable drops on perception tasks after integrating grounding (see Table 2). Rather than enhancing the model holistically, grounding behaves like an overhead that reduces overall perception capacity.

**Why the drop differs across methods.** We hypothesize two contributing factors: (i) *Data imbalance*: skewed ratios of VQA (perception) vs. referring/segmentation (grounding) data—especially in GLaMM—can induce catastrophic forgetting of perception skills; (ii) *Training scheme*: optimization tailored for grounding may overfit to segmentation objectives, eroding prior perception knowledge. In effect, the added grounding capability can overwrite existing perception competence, yielding a net regression in general visual understanding.

## 3 Look&Learn

Motivated by the aforementioned issues detailed in Section 2, we propose Look&Learn, an approach that encourages VLMs to attend to the correct image locations while generating responses, rather than explicitly performing a separate grounding task, which often leads to catastrophic forgetting of perception capabilities. To achieve this, we introduce a plug-and-play loss function, termed Where-to-Look, which can be seamlessly applied to the attention maps of any off-the-shelf VLM. This loss significantly enhances the model's grounding ability, mitigates hallucination, and improves understanding without requiring architectural modifications or additional modules.

### 3.1 Where-to-Look

As discussed in Section 2.1, existing VLM attention maps are sparse and scattered and often fail to focus on the correct image regions relevant to the conversation. Our goal is to guide an arbitrary VLM (composed of $L$ stacked attention layers) to direct its attention towards relevant visual cues at the correct time. In this section, we answer the following key questions: 1) When should we apply our loss? 2) How can we apply it efficiently?

### 3.1.1 Visual Entities Extraction

Our loss should only be applied when the model generates visual-related words, i.e., words that have a tangible visual counterpart in the image. Therefore, we categorize generated tokens into two groups: 1) Visual entities: Words corresponding to objects, actions, colors, spatial relationships, etc. (e.g., "eating", "red car", "sky"). 2) Linguistic words: Functional words with no direct visual representation (e.g., "is", "the", "I am"). To automatically extract visual entities, we use GPT-4o-mini to process the generated responses and extract key visual tokens. For example:

Figure 3: An overview of our **Where-to-Look loss**. The left side shows the overall pipeline: given an image, a system prompt, and a Q&A pair, the vision encoder and LLM generate attention maps ($M_{att}$) for each token. The highlighted visual entity (e.g., "cyan" shirt) is linked to its corresponding region in the attention map. The right side illustrates the sampling procedure: (1) obtain the attention map, (2) sample visual entities, (3) extract image-level attention, (4) process into a spatial map of size $h \times w$, and (5) compute cosine similarity between the predicted attention map and the pseudo mask $M_p$. This supervision guides the LLM to focus on the correct image regions.

> Q: What are the giraffes doing?
> A: The giraffes are eating from the tray.
> Extracted visual entities: ["giraffes", "eating", "tray"]

This ensures that the Where-to-Look loss is only applied when necessary, focusing on visual grounding entities without interfering with purely linguistic reasoning.

### 3.1.2 LOSS DESIGN

**Predicted Attention Map Extraction:** Given a Q&A conversation, we first extract:

- Visual entity tokens: $E_v \in \mathbb{R}^{N_E \times K}$, where $N_E$ is the number of extracted visual entities, and $K$ is the maximum token length per entity.
- Pseudo-ground truth masks: $M_p \in \mathbb{R}^{N_E \times H \times W}$, corresponding to the visual entities (see Section 3.2).

During training, the VLM naturally produces attention maps across all layers: $Att_{Map} \in \mathbb{R}^{L \times N \times N}$, where $N$ represents the total input sequence length (system prompt, image tokens, text tokens). To focus on the relevant attention signals, we extract attention only between the visual entities ($N_E$) in the output sequence and the image tokens ($N_{img}$) in the input sequence: $M_{att} \in \mathbb{R}^{L \times N_E \times K \times N_{img}}$ Since $N_{img}$ corresponds to flattened image patches (e.g., 576 tokens for CLIP-ViT L/16), we reshape them into a 2D attention map $(H', W')$, e.g., $H' = W' = 24$. To compare with the pseudo-ground truth masks $M_p$, we upsample $M_{att}$ to match the resolution $(H, W)$, resulting in: $M_{att} \in \mathbb{R}^{L \times N_E \times K \times H \times W}$

**Predicted Attention Map Aggregation:** To obtain a stable guidance signal, we aggregate $M_{att}$ across two axes:

- Across layers ($L$): Since it is difficult to determine the exact layer where grounding occurs, we take the average attention map across all layers.
- Across visual entity tokens ($K$): Since words like "eating" are tokenized into multiple subwords ("e", "ating"), we apply the loss to all corresponding tokens.

Thus, as shown in Figure 3, the final predicted and ground-truth masks are: $M_{att} \in \mathbb{R}^{N_E \times K \times H \times W}$, $M_p \in \mathbb{R}^{N_E \times K \times H \times W}$, where $M_p$ is repeated across the $K$ tokens.

Table 3: Statistics of our pseudo-mask collection pipeline across datasets. For each dataset, we report the number of image–text samples and extracted phrases, along with the fraction of phrases whose masks are (i) accepted by the *consensus* of four grounding models (**Consensus Pass**), (ii) additionally accepted by the LLM-based judge after consensus fails (**Judge Pass**), and (iii) rejected after both stages (**Rejected**). Rows in gray already provide bounding boxes, so SAM is directly applied to obtain masks (thus 100% Consensus Pass and no Judge Pass). For OCR-heavy datasets, we instead use the OCR-specific pipeline, so consensus/judge statistics are not applicable (N/A).

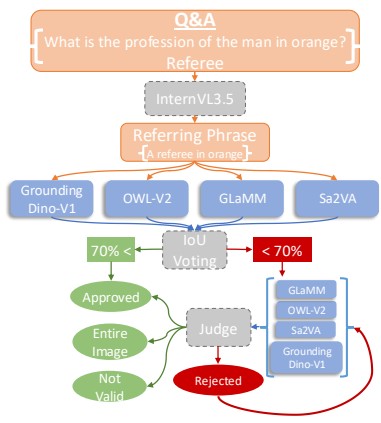

Figure 4: A detailed overview of our proposed pipeline for pseudo mask data collection.

| Dataset | # of Samples | # of Phrases | Consensus Pass | Judge Pass | Rejected |
|---|---|---|---|---|---|
| LLaVA-150K | 158K | 1.62M | 45% | 51% | 4% |
| VQA-v2 | 83K | 443K | 42% | 49% | 9% |
| OK-VQA | 9K | 9K | 51% | 44% | 5% |
| A-OK-VQA | 17K | 17K | 49% | 42% | 9% |
| OCR-VQA | 80K | 386K | N/A | N/A | N/A |
| TextCaps | 22K | 69K | 44% | 45% | 12% |
| GQA | 72K | 943K | 100% | 0% | 0% |
| VG | 86K | 835K | 100% | 0% | 0% |
| RefCOCO | 48.5K | 287.6K | 100% | 0% | 0% |

**Loss Formulation:** Since standard segmentation losses (e.g., cross-entropy, IoU loss) are inappropriate for our setting—where attention maps are probability distributions rather than binary masks—we require a scale-invariant loss. We propose using Cosine Similarity to align the predicted attention map $M_{att}$ with the pseudo-ground truth mask $M_p$:

$$\mathcal{L}_{\text{Where-to-Look}} = 1 - \frac{\sum_{i,j} M_{\text{att}}^{(i,j)} M_p^{(i,j)}}{\sqrt{\sum_{i,j}(M_{\text{att}}^{(i,j)})^2} \cdot \sqrt{\sum_{i,j}(M_p^{(i,j)})^2}} \tag{2}$$

where the loss encourages attention maps to match pseudo-ground-truth masks without enforcing absolute magnitudes, making it robust to soft attention distributions.

**Difference from Grounding Task:** Unlike traditional grounding tasks that require explicit mask generation or feature extraction, our loss function operates directly on the attention heatmap, ensuring that VLMs focus on the correct visual entities while generating corresponding text tokens. This approach maintains the original text-only output of VLMs without introducing a new modality, thereby preserving the model's inherent capabilities while improving grounding.

## 3.2 PSEUDO DATA COLLECTION PIPELINE

To supervise Where-to-Look, we require guidance signals that indicate where the model should focus while generating visual-related tokens. We therefore design a scalable pseudo-data collection pipeline that produces masks aligned with visual entities extracted from Q&A conversations (Fig. 4).

**Step 1: Referring phrase generation.** Given a Q&A pair, we instruct InternVL3.5 Wang et al. (2025) to compose a referring phrase for the relevant visual entity.

**Step 2: Multi-model grounding.** The phrase is passed through four diverse state-of-the-art grounding models: GLaMM Rasheed et al. (2024) and Sa2VA Yuan et al. (2025) (LLM-based grounding), OWL-v2 Minderer et al. (2023) and Grounding-DINO Liu et al. (2024b) (open-vocabulary segmentation). Combining LLM-based and segmentation-based methods ensures diversity and robustness.

**Step 3: Consensus voting.** If at least three of the four models produce masks with ≥70% IoU overlap, we accept the mask as the pseudo ground truth.

**Step 4: LLM judge.** If no consensus is reached, InternVL3.5 Wang et al. (2025) acts as a judge: each candidate mask is sequentially overlaid on the image and classified as *Approved*, *Entire Image*, *Not Valid*, or *Rejected*. The order of evaluation is determined by the models' accuracy on LVIS (Table 11). The process continues until a valid mask is found or all candidates are rejected, in which case the sample is discarded.

**Scalability.** The pipeline is modular and can integrate new grounding models seamlessly, ensuring adaptability and improved pseudo-mask quality over time.

Table 4: A comparison between our approach and LlaVA on the reasoning VQA benchmarks. Ours-Ents and Ours-Seg indicate where our loss is applied to the visual entities or the SEG token.

| Method | VQA-v2 | GQA | SQA | POPE | TextQA | MMVet | RefCOCOg | LlaVA-Bench$hW$ | Count-Bench |
|---|---|---|---|---|---|---|---|---|---|
| LlaVA | 78.5 | 62.0 | 66.8 | 85.9 | **58.0** | 30.5 | 61.4 | 63.4 | 43.2 |
| LlaVA-G | 75.1 | 58.3 | 64.0 | 85.1 | 54.4 | 25.5 | 68.4 | 57.9 | - |
| GLaMM | 18.5 | 14.6 | 40.4 | 60.7 | 8.4 | 6.7 | 20.4 | 23.6 | - |
| **Ours-Ents** | **79.2** | **63.6** | **68.4** | **86.7** | 57.7 | **31.5** | 65.5 | **66.9** | **45.7** |
| **Ours-Seg** | **78.9** | 62.9 | 67.2 | 86.6 | 56.3 | 31.0 | 67.3 | 64.7 | 45.2 |

**Statistics.** As summarized in Table 3, we collect ∼2.5M visual entities with corresponding masks. For datasets already providing bounding boxes (highlighted in gray), we directly apply SAM to generate masks.

**OCR Special Handling.** Grounding models are often unreliable for text-in-the-wild scenarios (e.g., book covers, posters), where visual entities correspond to words rather than objects. To handle such cases, we design a dedicated OCR-aware pipeline. Specifically, we apply TextSnake (Long et al., 2018) to detect curved and irregular text regions, followed by SVTR-v2 (Du et al., 2025) for robust text recognition. Masks are then generated directly from the detected text regions. Qualitative examples are provided in Fig. 5.

**Human Evaluation.** To validate the quality of our pseudo-masks, we conduct a human evaluation on 10K samples drawn from all datasets in Table 3. Annotators use a lightweight interface (Fig. 6) that displays SAM-generated mask proposals and allows quick selection or rejection based on the referring phrase. This setup reduces annotation time by roughly 90% compared to drawing masks manually. Overall, 93% of our pseudo-masks are confirmed to be correct. Additional implementation details and the evaluation protocol are provided in Appendix B.2.

## 4 EXPERIMENTS

**Overview.** We evaluate Look&Learn in two complementary settings that mirror our core claims (Mitigating the understanding and grounding gap, LLM as Segmentor, plug-and-play, phase-agnostic). **Part A (Mitigating understanding and grounding gap).** We integrate Where-to-Look loss into the standard two-stage LLaVA-1.5 pipeline and train with paired *conversations + pseudo masks* collected by our pipeline (Sec. 3.2). This tests whether aligning attention during fine-tuning improves both grounding and understanding under the original LLaVA recipe. **Part B (LLM as segmentor).** We drop in our loss, *without any new data*, into GLaMM to isolate the contribution of the loss itself. We report two modes: (i) fine-tuning with Where-to-Look loss and (ii) *post-training* (no re-finetuning), demonstrating phase-agnostic deployment.

### 4.1 PART A: MITIGATING UNDERSTANDING AND GROUNDING GAP

**Training data.** For a fair comparison, we use exactly the training datasets of LLaVA-1.5 and augment them with pseudo masks aligned to visual entities in the conversations (Table 3). This provides paired supervision—*conversation turns + entity masks*—without requiring GT segmentation.

**Training strategy.** We follow LLaVA's two-stage recipe: *Stage 1 (pre-training)*: freeze the LLM, train only the projection layer *without* attention guidance to establish multimodal alignment. *Stage 2 (fine-tuning)*: enable Where-to-Look on cross-modal attention, updating the projection and LLM parameters jointly. All other hyperparameters match LLaVA for a controlled comparison.

**Evaluation protocol.** We assess both *understanding* and *grounding* to quantify the perception–grounding tradeoff: (i) **Understanding**: VQA-v2, GQA, SQA, POPE, TextQA, MM-Vet, LLaVA-Bench, Counting-Bench. (ii) **Grounding**: RefCOCO, RefCOCO+, RefCOCOg (referring segmentation). This joint evaluation measures whether attention guidance narrows the gap while preserving (or improving) understanding.

### 4.1.1 LLAVA'S RESULTS ON UNDERSTANDING BENCHMARKS

Where-to-Look consistently improves reasoning while reducing hallucination. On POPE, we observe stronger hallucination robustness; across VQA-style tasks, we obtain ∼1–2% average

Table 5: Evaluating the plug-and-play nature of our Where-to-Look loss on Qwen2.5VL-3B and Qwen3VL-4B. "N/A" denotes the original pretrained model without any fine-tuning. ✗ and ✓ indicate fine-tuning without and with our loss, respectively.

| Model | Our Loss | RefCOCO | | | RefCOCO+ | | | RefCOCOg | |
|---|---|---|---|---|---|---|---|---|---|
| | | Val | Test A | Test B | Val | Test A | Test B | Val | Test |
| Qwen2.5VL-3B | N/A | 51.5 | 52.4 | 48.8 | 41.1 | 42.4 | 38.9 | 48.2 | 49.7 |
| | ✗ | 54.0 | 60.1 | 54.8 | 44.2 | 47.2 | 41.9 | 49.8 | 50.9 |
| | ✓ | **60.2** | **61.6** | **57.5** | **47.2** | **49.0** | **44.9** | **53.6** | **54.9** |
| Qwen3VL-4B | N/A | 65.1 | 68.3 | 60.3 | 55.0 | 62.2 | 48.6 | 66.5 | 66.2 |
| | ✗ | 66.6 | 70.3 | 62.5 | 56.7 | 61.5 | 52.1 | 61.2 | 61.7 |
| | ✓ | **72.9** | **75.8** | **68.0** | **64.9** | **70.0** | **58.7** | **68.9** | **69.4** |

gains over LLaVA-1.5 under identical data and hyperparameters (Table 4). Compared to LLaVA-G—which uses more datasets—our model remains competitive or better on most reasoning benchmarks, indicating that attention alignment benefits understanding rather than trading it off.

### 4.1.2 SCALING TO STRONGER VLMS: QWEN2.5VL AND QWEN3VL

To demonstrate that Where-to-Look is architecture-agnostic and easily integrates into modern VLMs, we apply our loss to two state-of-the-art models: Qwen2.5VL and Qwen3VL. Starting from their released pretrained checkpoints, we perform a very light LoRA fine-tuning for only one epoch, using the same training setup, learning rate, and LoRA configuration for all variants. For each model, we fine-tune twice: once without our loss (vanilla fine-tuning) and once with our loss, ensuring a fair, controlled comparison. As shown in Table 5, our loss consistently improves performance across all benchmarks. Specifically, on Qwen2.5VL-3B, we obtain gains of +5.8 / +7.2 / +8.7 on RefCOCO (Val / TestA / TestB), +5.1 / +7.0 / +2.8 on RefCOCO+, and +4.8 / +4.0 on RefCOCOg (Val / Test). On the stronger Qwen3VL-4B, our loss yields +8.0 / +7.0 / +5.4 on RefCOCO, +8.2 / +8.5 / +6.6 on RefCOCO+, and +7.7 / +7.7 on RefCOCOg. These results confirm that our loss is plug-and-play, lightweight, and highly effective even for strong modern VLMs.

**An interesting trend emerges across architectures:** our gains increase as the underlying vision backbone becomes more expressive. On LLaVA, which uses a relatively modest ViT encoder, we observe only 1–2% improvements. On Qwen2.5VL-3B, the average gain rises to $\approx 5.8\%$, and on the stronger Qwen3VL-4B, equipped with a significantly more capable visual stack, the average gain further increases to $\approx 7.7\%$. We hypothesize that our attention-guidance loss benefits more from richer visual features, as the LLM can more effectively align its internal attention to fine-grained spatial signals. Qwen3VL introduces several visual enhancements that make this alignment particularly effective: (i) Interleaved-MRoPE, which distributes spatial and temporal dimensions across all frequency bands, yielding more robust positional grounding; and (ii) DeepStack multi-level feature fusion, which injects visual tokens from multiple ViT layers into multiple LLM layers, preserving both low- and high-level visual cues. These improvements provide stronger, more detailed visual signals for our loss to exploit, explaining the larger performance gains.

### 4.1.3 RESULTS ON REFERRING SEGMENTATION

**Text-predicted boxes → masks.** Replacing only the loss (our method vs. vanilla LLaVA) yields ∼3% absolute gains across RefCOCO/+/g when text-predicted boxes are converted to masks (Table 6, top).

**Native attention masks (LLM as Segmentor).** Using attention maps directly as masks, we outperform GLaMM by 4–6% despite GLaMM's reliance on SAM and GT masks (Table 6, middle), supporting the *LLM-as-Segmentor* premise.

**Zero-shot SAM refinement.** As a post-process, converting our attention to boxes or keypoints and feeding them to SAM boosts performance by 8–10% without training (Sec. C.2). While optional, this shows that our attention is a strong, composable prior for classical segmentors.

**On fairness.** Our model never sees GT masks for these datasets; nevertheless, it matches or exceeds methods fine-tuned on them—highlighting that attention-guided grounding can bridge the perception–grounding gap without bespoke decoders or extra parameters.

Finally, by finetuning both the VLM (LLaVA) and the SAM, similar to GLaMM, we achieve better performance than GLaMM by more than 1% (Table 6, bottom).

Table 6: Comparison of our method against segmentation-based VLMs.

| Model | Output | Seg. Dec. | GT Seg. Data | Finetuning Seg. Dec. | RefCOCO | | | RefCOCO+ | | | RefCOCOg | |
|---|---|---|---|---|---|---|---|---|---|---|---|---|
| | | | | | Val | Test A | Test B | Val | Test A | Test B | Val | Test |
| LlaVA | BBox | ✗ | ✗ | ✗ | 43.5 | 44.8 | 41.6 | 35.2 | 37.6 | 31.6 | 43.7 | 44.1 |
| **Ours** | BBox | ✗ | ✗ | ✗ | **45.8** | **47.4** | **44.9** | **37.8** | **40.1** | **35.1** | **46.0** | **47.5** |
| GLaMM | Mask | ✓ | ✓ | ✗ | 54.7 | 58.1 | 52.2 | 42.5 | 47.1 | 39.5 | 54.8 | 55.6 |
| **Ours** | Att. Map | ✗ | ✗ | ✗ | **60.2** | **62.9** | **58.7** | **49.0** | **54.4** | **46.1** | **57.7** | **58.7** |
| **Ours** | Mask | ✓ | ✗ | ✗ | **70.0** | **73.7** | **69.1** | **58.1** | **65.8** | **55.0** | **69.4** | **70.6** |
| LlaVA-G | Mask | ✓ | ✓ | ✓ | 77.13 | - | - | 68.8 | - | - | 71.5 | - |
| LISA | Mask | ✓ | ✓ | ✓ | 74.9 | 79.1 | 72.3 | 65.1 | 70.8 | 58.1 | 67.9 | 70.6 |
| GSVA | Mask | ✓ | ✓ | ✓ | 77.7 | 79.9 | 74.2 | 68.0 | 71.5 | 61.5 | 73.2 | 73.9 |
| GLaMM | Mask | ✓ | ✓ | ✓ | 79.5 | 83.2 | 76.9 | 72.6 | 78.7 | 64.6 | 74.2 | 74.9 |
| **Ours** | Mask | ✓ | ✗ | ✓ | 74.6 | 79.2 | 74.9 | 65.3 | 71.8 | 61.0 | 73.1 | 72.4 |
| **Ours** | Mask | ✓ | ✓ | ✓ | **80.8** | **83.9** | **77.8** | **73.8** | **79.5** | **65.8** | **74.9** | **75.7** |

Table 7: Integrating our loss into GLaMM under fine-tuning (FT) and post-training (PT).

| Model | Training Stage | RefCOCO | | | RefCOCO+ | | | RefCOCOg | |
|---|---|---|---|---|---|---|---|---|---|
| | | Val | Test A | Test B | Val | Test A | Test B | Val | Test |
| GLaMM | FT | 78.65 | 81.08 | 76.66 | 70.89 | 76.14 | 61.99 | 74.26 | 75.71 |
| **Ours** | FT | **79.73** | **82.34** | **77.12** | **70.99** | **77.36** | **64.77** | **74.22** | **75.64** |
| **Ours** | PT | **80.42** | **83.52** | 76.45 | **72.0** | **76.29** | **65.27** | **74.79** | **75.9** |

## 4.2 PART B: LLM AS SEGMENTOR

**Plug-and-play evaluation on GLaMM.** To disentangle the effect of additional pseudo masks from the contribution of Where-to-Look, we integrate our loss directly into GLaMM *without introducing new data*. This isolates the loss itself as the only change, making the setup fair and controlled.

**Two modes.** We test two integration modes: (i) *Fine-tuning (FT)* where Where-to-Look is added during training, and (ii) *Post-training (PT)* where the loss is applied after fine-tuning, requiring no retraining of the backbone. PT is especially attractive for practitioners since it improves grounding without costly re-training.

**Results.** Table 7 shows consistent gains across RefCOCO, RefCOCO+, and RefCOCOg in both modes. Notably, even in the PT setup, our method outperforms the original GLaMM. These results validate that Where-to-Look is a *true plug-and-play mechanism*, enabling any VLM to act as a segmentor without architectural changes, additional data, or full retraining. Qualitative results could be seen in Appendix C.4.

## 5 DISCUSSION AND CONCLUSION

**LLM as Segmentor.** Our results demonstrate that, with Where-to-Look, the LLM itself can act as a segmentor by producing high-quality attention maps that align with visual entities. This opens the door to replacing heavy segmentation decoders such as SAM (Lin et al., 2025) with lightweight mask refiners (Wang et al., 2023a) that simply sharpen the attention-derived masks. Such a design is attractive both for training—removing the need for costly segmentation supervision—and for inference, where it reduces memory and compute overhead. This efficiency makes large-scale deployment of grounded VLMs more practical.

**Attention-guided multi-turn inference.** Another promising direction is to leverage our high-quality segmentation to *improve understanding itself*. Specifically, a VLM can first generate an initial answer alongside its attention maps; these maps are then converted into masks of the relevant regions. By feeding the masked region back into the model as an auxiliary input (together with the original image and question), the model can refine its reasoning in a second turn. This multi-turn, attention-guided setup effectively uses the model's own grounding to sharpen its understanding, offering a path toward more reliable and interpretable inference.

**Conclusion.** In summary, Look&Learn provides a simple yet powerful mechanism for unifying perception and grounding. By reframing the LLM as a segmentor, our approach avoids the traditional understanding–grounding tradeoff, delivering gains on both. Beyond strong empirical results, the potential applications outlined above suggest broader impacts: more efficient architectures, better reasoning through self-grounding, and a step toward interpretable multimodal systems. We hope this work inspires further exploration of attention-guided learning as a foundation for the next generation of VLMs.

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

# A    APPENDIX: RELATED WORK

Recent work has explored the use of attention maps from vision–language models for segmentation. Luo et al. (2024) shows that CLIP's cross-attention layers contain spatial cues and uses a training-free procedure to extract zero-shot segmentation masks. However, PnP-OVSS operates purely on CLIP encoders, does not involve LLMs, and cannot influence multimodal reasoning or grounding during text generation. MaskCLIP Dong et al. (2023) and MaskCLIP++ Zeng et al. (2024) enhance CLIP pretraining by incorporating masked self-distillation to improve local patch representations; however, they neither generate segmentation masks nor supervise the LLM's attention. In contrast, our method directly guides LLM attention during generation, enabling the model itself—rather than a CLIP encoder or an external decoder—to produce high-quality segmentation masks. End-to-end grounding models such as MDETR Kamath et al. (2021) and the Referring Transformer Li & Sigal (2021) jointly reason over text and image features by adding task-specific detection or segmentation heads that directly predict boxes and masks for expressions. While highly effective, these multi-task methods rely on explicit grounding supervision and specialized decoders, and do not operate on or modify the LLM's internal attention mechanisms. Another line of work uses attention supervision to mitigate language priors. HINT Selvaraju et al. (2019) aligns model importance with human attention maps for VQA and captioning, but it requires human annotations, does not produce segmentation masks, and does not provide a mechanism for unifying grounding with text generation in instruction-tuned VLMs.

A large body of work has explored guiding diffusion models using spatial masks or attention-based constraints to improve controllability and semantic alignment during image generation. Mask-guided approaches such as DiffEdit Couairon et al. (2022), InstructEdit Wang et al. (2023b), TWIG Rakib et al. (2025), and Panoptic Diffusion Long & Roy (2024) rely on explicit segmentation masks to localize edits, preserve structure, or control object placement throughout the denoising process. Complementary to these, attention-based methods—including Prompt-to-Prompt Hertz et al. (2022), Attend-and-Excite Chefer et al. (2023), Loss-Guided Diffusion Song et al. (2023), regional or temporal adaptive attention map control, and various self-attention guidance strategies—manipulate or supervise the model's cross- and self-attention maps to enforce stronger semantic grounding or reduce attention drift. While these methods demonstrate that both masks and attention maps are powerful conditioning signals for diffusion, they predominantly operate in the image-generation domain and focus on heuristic or inference-time control.

Recent studies on the limitations and internal mechanisms of multimodal LLMs highlight the importance of attention quality and localization cues for reliable visual understanding. Towards Perceiving Small Visual Details in Zero-shot VQA Zhong et al. (2024) demonstrates that MLLMs struggle substantially with small visual subjects, and shows that performance can be recovered through human or automatic visual cropping. This underscores that current MLLMs are highly sensitive to spatial detail and often fail to attend to the correct fine-grained regions of the image. Complementary findings are reported in Your Large Vision-Language Model Only Needs a Few Attention Heads for Visual Grounding Kang et al. (2025), where the authors reveal that only a small subset of cross-attention heads—termed localization heads—naturally capture object-level grounding signals without any fine-tuning. Leveraging these heads enables strong training-free grounding performance, suggesting that VLMs possess latent grounding capabilities but do not consistently activate them across tasks. Together, these works highlight a growing consensus: the spatial selectivity and attention allocation of MLLMs are crucial bottlenecks, and improving how models attend to fine-grained visual regions is central to advancing VQA and grounding performance—motivating our focus on explicit attention guidance during training.

Unlike these approaches, our method does not introduce any detection or segmentation module and does not require human attention labels; instead, we directly guide LLM attention during generation using pseudo masks, allowing the model to act as its own segmentor while simultaneously addressing the understanding–grounding trade-off that prior grounding or attention-supervised methods do not investigate. More importantly, none of the existing methods addresses the understanding–grounding gap: the degradation in perception that occurs when segmentation modules are added to VLMs. Our work is the first to identify this phenomenon and show that attention guidance inside the LLM can simultaneously improve grounding and preserve (or even enhance) high-level understanding.

## B APPENDIX: DATA CURATION PIPELINE; MORE DETAILS

### B.1 OCR SPECIAL HANDLING

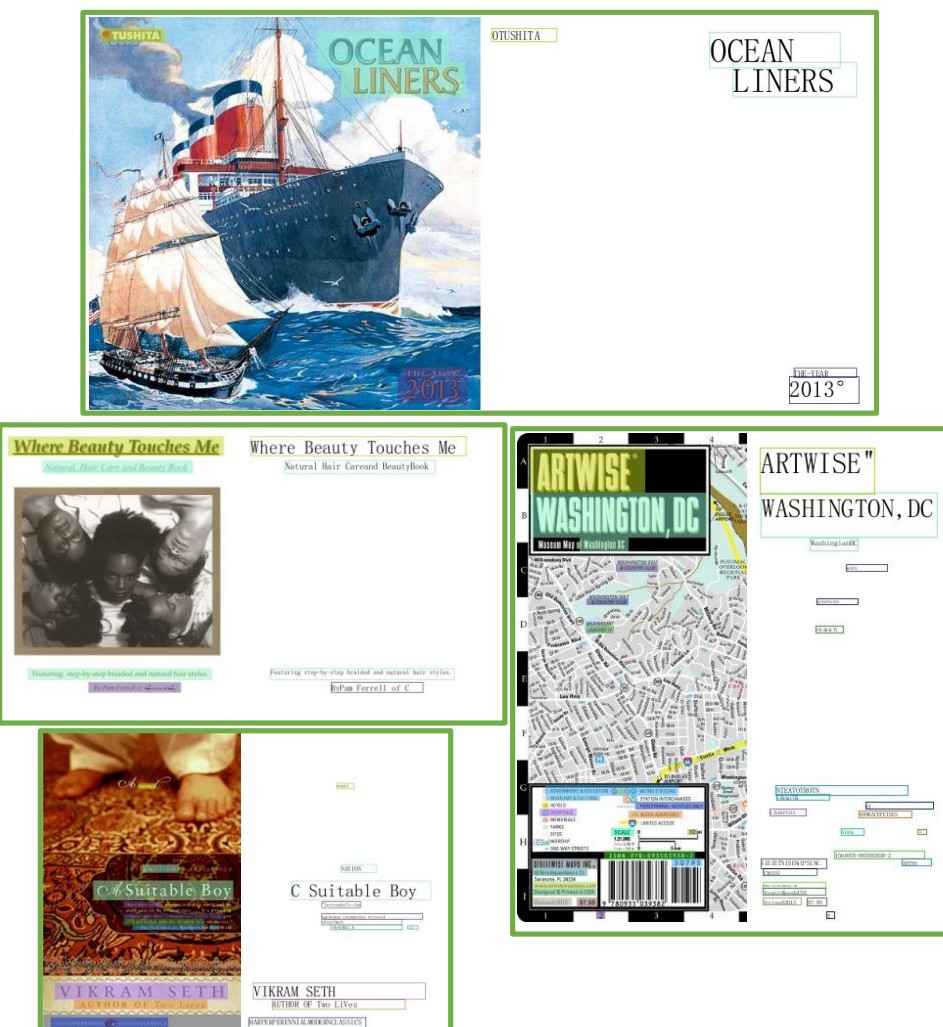

Figure 5: Qualitative results from our OCR-specific data pipeline. For text-rich images (e.g., book covers and posters), TextSnake detects irregular and curved text regions, and SVTR-v2 recognizes their content. The detected text regions (right) serve as pseudo-masks aligned with the original images (left), enabling reliable supervision for grounding in OCR-heavy scenarios.

### B.2 HUMAN EVALUATION

To assess the reliability of our data collection pipeline, we conduct a human evaluation study over 10K sampled masks drawn from the datasets listed in Table 3. To streamline annotation, we designed a lightweight GUI (Fig. 6). For each sample, we first extract all candidate masks using SAM, without conditioning on the referring expression. The annotator is then presented with the referring phrase and asked to choose among four options:

- Select the correct mask if one of the SAM proposals matches the described region;
- Entire Image if the expression is globally descriptive (e.g., "black and white image");
- Can't Ground if the phrase is vague or invalid;

- No Mask if SAM fails to propose an appropriate mask—for example, when the expression refers to a fine-grained part of an object but SAM only detects the whole object.

This interface reduces annotation time by approximately 90% compared to drawing masks from scratch. Overall, the study shows that 93% of our pseudo-masks are validated as correct, confirming the reliability of our pipeline.

### B.3 RUNTIME ANALYSIS OF THE DATA COLLECTION PIPELINE

The scalability of the data pipeline is crucial for its practical adoption. Our pipeline is designed to be lightweight, parallelizable, and fully open-source. Using a single node equipped with $8\times$A100 GPUs, we can process the entire 2.5M visual entities in approximately one day, as the four grounding models run concurrently on separate GPUs and the judge model operates in parallel over only half of the samples. InternVL3.5—used for both referring-phrase generation and the judging stage—is served via the vLLM inference engine, achieving a $6\times$ speedup compared to standard HuggingFace/Transformers inference, which further reduces latency. A detailed breakdown of the per-module throughput and estimated runtime is provided in Table 8.

Table 8: Estimated throughput and runtime for each component of the data collection pipeline. Times are measured using a single A100 GPU per model; in practice, the pipeline runs all grounding models in parallel on an $8\times$A100 node, resulting in a total wall-clock time of roughly one day for processing 2.5M samples.

| Model | Task | Throughput | Time for 2.5M Samples |
|---|---|---|---|
| Sa2VA | Segmentation grounding | $\sim$5.6 samples/sec | $\sim$125 hours |
| GLaMM | Segmentation grounding | $\sim$22 samples/sec | $\sim$31 hours |
| OWL-V2 | OV grounding | $\sim$66.6 samples/sec | $\sim$10.4 hours |
| Grounding DINO | OV grounding | $\sim$66.6 samples/sec | $\sim$10.4 hours |
| InternVL3.5 (Judge) | Evaluate 50% of samples | $\sim$13.9 samples/sec | $\sim$25 hours |
| InternVL3.5 (Phrase) | Referring-phrase generation | $\sim$41.7 samples/sec | $\sim$16.7 hours |

In summary, our updated pipeline is fast, reproducible, and feasible for any lab with access to a single multi-GPU node, making it considerably simpler and more efficient than prior multi-stage grounding pipelines such as GLaMM.

## C APPENDIX: MORE EXPERIMENTS

### C.1 ABLATIONS

**Layer aggregation** ($L$). Applying Where-to-Look at a single hand-picked layer (e.g., mid or late) is competitive, but enforcing it *at all layers* hurts performance (rigid constraints). *Averaging across layers* achieves the best balance, letting multiple layers contribute while retaining reasoning capacity (Table 9a).

**Visual-token aggregation** ($K$). Our ablations show that *averaging* across all visual tokens provides the most stable and effective grounding signal, as attention maps are inherently fragile and supervising each token independently introduces noisy gradients and degrades performance (Table 9b).

**Loss interplay (next-token vs. attention guidance).** When supervising *visual entities directly*, performance is sensitive to the weight between cross-entropy and Where-to-Look due to shared heatmaps. Using a dedicated ⟨SEG⟩ token decouples objectives and is less sensitive to weighting, but direct entity supervision yields larger gains when tuned properly (Table 9c).

**Do We Need the ⟨SEG⟩ Token?** The role of the ⟨SEG⟩ token in our method is fundamentally different from how it is used in prior segmentation-based VLMs (e.g., GLaMM, Sa2VA, LISA). In those models, the ⟨SEG⟩ token is a semantic container: the segmentation mask is encoded in the hidden state of the token, and an external decoder (typically SAM) must be trained to decode this hidden representation back into a fine-grained mask. In contrast, our method does not use the hidden-state features of the ⟨SEG⟩ token at all. All mask information comes exclusively from the attention

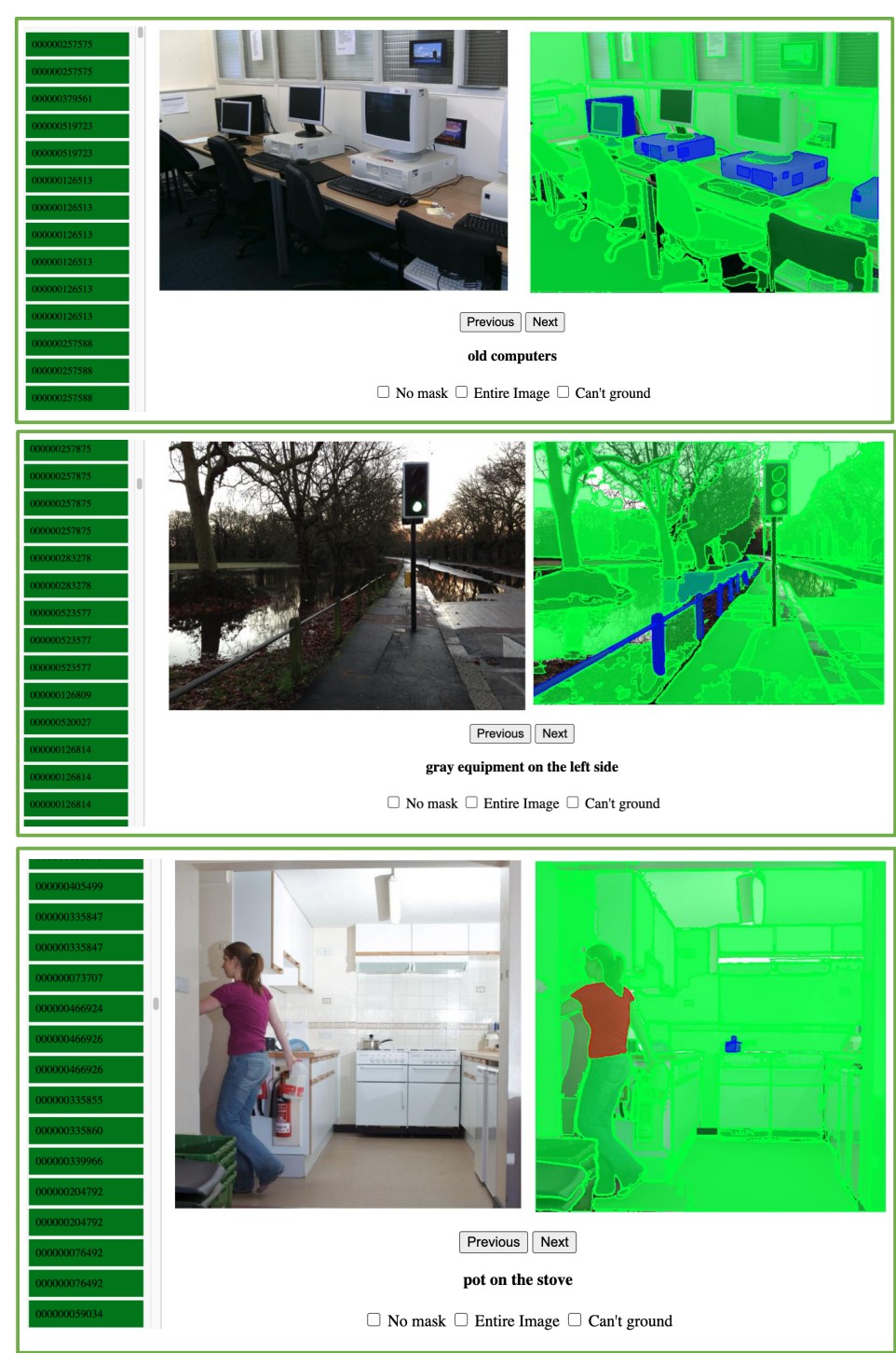

Figure 6: Human evaluation interface used to validate 10K pseudo-masks. The left panel lists sample IDs; the center shows the original image; and the right panel displays all unconditional SAM-generated masks (in green). When the annotator hovers over a candidate mask, it is highlighted in red, and the currently selected mask is shown in blue. The tool also provides annotation options ("No mask", "Entire Image", "Can't Ground") to efficiently assess mask correctness and grounding validity.

Table 9: **Ablation study of the Where-to-Look loss.** We analyze four key factors affecting the effectiveness of our attention-guidance loss. (a) *Aggregation across layers (L):* averaging supervision across all layers yields the strongest results, confirming that distributing the signal stabilizes training. (b) *Aggregation across visual tokens (K):* averaging across tokens provides the best grounding performance, as attention maps are fragile and independent-token supervision introduces noise. (c) *Loss weighting:* entity-based supervision (Ents) and segmentation-based supervision (SEG) behave differently depending on the relative weighting of text (Next-Token-Prediction)- and our loss. (d) *Effect of mask quality:* our pseudo-grounding pipeline provides mask quality nearly on par with SAM while outperforming Grounding-DINO and GT bounding box conversion. All models are trained on VQA-v2 and evaluated on the indicated benchmarks.

(a) Aggregation across $L$

| Method | VQA-v2 (Val) |
|---|---|
| **L=31** | 49.8 |
| **L=15** | 49.2 |
| **All** | 48.3 |
| **Max** | 49.9 |
| **Avg.** | **50.5** |

(b) Aggregation across $K$

| Method | VQA-v2 (Val) |
|---|---|
| **All** | 47.8 |
| **Max** | 49.4 |
| **Avg.** | **50.0** |

(c) Text vs. our loss weighting

| Method | VQA-v2 (Val) | RefCOCOg |
|---|---|---|
| **Ents (1:1)** | 49.1 | 66.7 |
| **Ents (10:1)** | 50.5 | 64.3 |
| **SEG (1:1)** | 50.1 | 66.8 |
| **SEG (10:1)** | 50.1 | 66.5 |

(d) Effect of mask quality

| Mask Source | GQA |
|---|---|
| **GT-bbox** | 60.1 |
| **SAM masks** | **62.8** |
| **Grounding-Dino** | 60.4 |
| **Our Pipeline** | 62.7 |

Table 10: curriculum scheduling of attention guidance ablation study using Qwen3VL-4B. "N/A" denotes the original pretrained model without any fine-tuning. ✗ and ✓ indicate fine-tuning with our loss with and without curriculum scheduling, respectively.

| Model | Curriculum Scheduling | RefCOCO | | | RefCOCO+ | | | RefCOCOg | |
|---|---|---|---|---|---|---|---|---|---|
| | | Val | Test A | Test B | Val | Test A | Test B | Val | Test |
| **Qwen3VL-4B** | N/A | 65.1 | 68.3 | 60.3 | 55.0 | 62.2 | 48.6 | 66.5 | 66.2 |
| | ✗ | 70.3 | 72.4 | 65.3 | 61.5 | 65.6 | 57.1 | 65.7 | 66.8 |
| | ✓ | **72.9** | **75.8** | **68.0** | **64.9** | **70.0** | **58.7** | **68.9** | **69.4** |

map associated with the token, not from its features. This distinction is central: our approach is attention-native, not decoder-based, and therefore does not require any special segmentation token by design. A normal visual word (e.g., "car") already produces an attention heatmap that our loss can supervise. When we add a ⟨SEG⟩ token in the Ours-Seg variant, it serves only as a convenience anchor for grouping visual tokens during inference—not as a mechanism for encoding masks in its hidden state. The segmentation still emerges purely from its attention map, which is guided by our loss exactly like any other visual token.

**Quality of pseudo masks.** On GQA (with GT boxes), SAM-converted masks and our full multi-model pipeline provide the most reliable guidance; GT boxes alone or single-model pseudo masks are weaker (Table 9d).

**Curriculum Scheduling of Attention Guidance.** Attention guidance can be fragile or even harmful if applied naively; therefore, we conducted an additional curriculum-style ablation on the weighting of our loss. Instead of using a fixed ratio between the next-token prediction loss and our Where-to-Look loss, we adopt a three-stage schedule: during the first third of training, both losses are weighted equally (1:1); during the second third, the weight of our loss is linearly decayed to zero; and during the final third, training proceeds using only the standard next-token prediction loss. This curriculum strategy consistently improves performance by 2-3% on average across benchmarks compared to fixed-weight supervision (e.g., 1:1 or 2:1). This experiment yields three important insights: (1) Aggressive or prolonged attention supervision can indeed be harmful, confirming prior concerns in the literature and highlighting the necessity of careful, staged application. (2) Right attention does not necessarily imply right reasoning: although both fixed and scheduled variants produce similarly aligned attention heatmaps, only the flexible curriculum improves downstream accuracy—empirically reinforcing the view that attention is not itself the explanation. (3) Grounding signals persist even after guidance is removed: despite completely disabling our loss in the final third of training, the attention maps remain sharp and well-aligned, indicating that the network internalizes grounding as a useful inductive bias rather than a constraint imposed by a continuous penalty. This directly supports our central "unsolicited advice" hypothesis: when grounding is harmonized with the main objective, the model retains and exploits it naturally, rather than overwriting it once supervision is removed. (Table 10).

## C.2 SAM ZERO SHOT SAMPLING STRATEGIES

To refine our native segmentation masks and improve their quality, we extract key points from the attention maps and use them as guidance for downstream segmentation models (e.g., SAM). Our key-point selection strategy ensures that we obtain both high-confidence positive points (pixels that strongly correspond to the visual entity) and negative points (background pixels that should not be considered part of the segmentation mask).

### C.2.1 KEY-POINT SELECTION STRATEGY

To refine our native segmentation masks and improve their quality, we extract key points from the attention maps and use them as guidance for downstream segmentation models (e.g., SAM). Our key-point selection strategy ensures that we obtain both high-confidence positive points (pixels that strongly correspond to the visual entity) and negative points (background pixels that should not be considered part of the segmentation mask).

### C.2.2 KEY-POINT SELECTION STRATEGY

Given an attention map $A \in \mathbb{R}^{H \times W}$, where each pixel value represents a probability distribution over the image space, we sample two types of key points:

1. Positive Key Points ($\mathcal{P}_{\text{pos}}$) – Pixels that have high attention values and are most likely to belong to the target object. 2. Negative Key Points ($\mathcal{P}_{\text{neg}}$) – Pixels with low attention values, ensuring background supervision.

### C.2.3 POSITIVE KEY-POINT SAMPLING

To select positive key points, we apply a threshold-based filtering mechanism:

$$\mathcal{P}_{\text{pos}} = \{(i, j) \mid A(i, j) \geq \tau_{\text{pos}}\}$$

where $\tau_{\text{pos}}$ is a pre-defined threshold (e.g., 0.5), ensuring that only pixels with high attention probability are selected.

Since not all pixels contribute equally, we sample points proportionally to their attention scores, using a weighted probability distribution:

$$P(i, j) = \frac{A(i, j)}{\sum\limits_{(i', j') \in \mathcal{P}_{\text{pos}}} A(i', j')}$$

where $P(i, j)$ represents the sampling probability of pixel $(i, j)$ among the selected high-attention pixels. Using this probability, we sample $N_{\text{pos}}$ positive points.

If the number of positive candidate pixels is less than $N_{\text{pos}}$, we apply sampling with replacement to ensure sufficient positive points.

### C.2.4 NEGATIVE KEY-POINT SAMPLING

To obtain negative key points, we consider pixels with low attention values:

$$\mathcal{P}_{\text{neg}} = \{(i, j) \mid A(i, j) < \tau_{\text{neg}}\}$$

where $\tau_{\text{neg}}$ is a lower threshold (e.g., 0.2), ensuring that only background pixels are selected. Unlike positive key-point selection, negative points are sampled uniformly from the candidate set:

$$P(i, j) = \frac{1}{|\mathcal{P}_{\text{neg}}|}$$

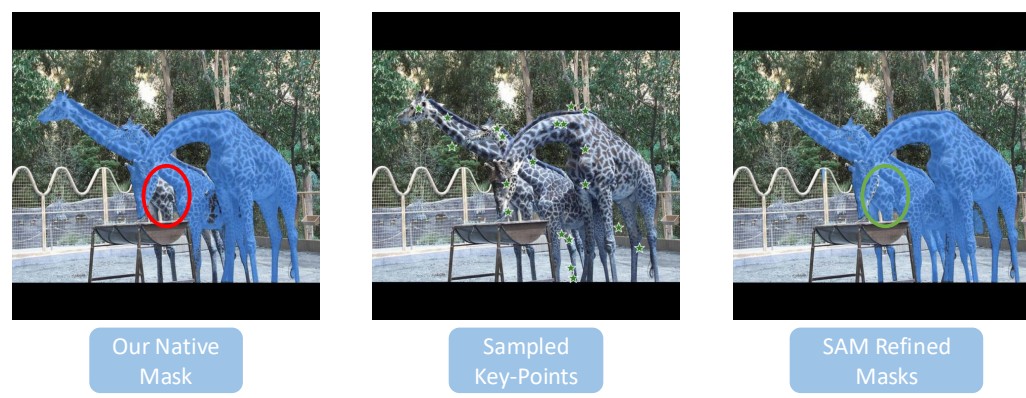

Figure 7: A qualitative example for the key-points sampling for SAM mask refinement.

Table 11: Benchamrk different grounding models on LVIS.

| Model | LVIS (Cls Name) | LVIS (Cls Def). |
|---|---|---|
| Grounding-Dino | 39.5 | 34.39 |
| OWL-v2 | 51.1 | 32.27 |
| Sa2VA | 42.2 | 32.53 |
| GLaMM | **51.1** | **40.8** |

Where all negative candidates have an equal probability of being selected, we sample $N_{\text{neg}}$ negative key points, ensuring a diverse background selection.

### C.2.5 IMPLEMENTATION DETAILS

Given an input attention map $A \in \mathbb{R}^{H \times W}$, our algorithm proceeds as follows:

1. Flatten the attention map into a 1D array. 2. Extract positive candidates $\mathcal{P}_{\text{pos}}$ and sample based on attention-weighted probabilities. 3. Extract negative candidates $\mathcal{P}_{\text{neg}}$ and sample uniformly. 4. Convert sampled indices back to (x, y) coordinates in the image space.

### C.3 LVIS BENCHMARK

To rank the models used in our data collection pipeline—Sa2VA, GLaMM, Grounding-DINO, and OWL-v2—we evaluate their performance on the LVIS benchmark. As shown in Table 11, GLaMM achieves the highest performance, followed by OWL-v2, Sa2VA, and Grounding-DINO in descending order. Based on these results, we adopt this ranking as the processing order in our pipeline (Figure 4 of main paper) to maximize the quality of the generated pseudo masks.

### C.4 QUALITATIVE EXAMPLES

We present three types of qualitative results to illustrate the effectiveness of our approach: **(i) Grounded-based Understanding**, where we visualize the model's predicted answers alongside the corresponding attention maps; **(ii) LLM as Segmentor**, where attention maps are directly used as segmentation masks after training on referring datasets; and **(iii) Output Masks with SAM Refinement**, where we show the final refined masks after integrating our loss into GLaMM with SAM.

**Grounded-based Understanding.** Figure 8 shows examples where our model generates both accurate answers and aligned attention maps. For instance, when asked "What is the direction of cars on the side of the road?", the model correctly answers "Forward" while its attention highlights the cars. Similarly, for "Is there a driver inside the bus?", the answer "Yes" is supported by attention focused on the bus window. These examples demonstrate that our loss encourages the LLM to attend to the correct image regions while producing answers, reducing spurious attention to irrelevant areas and grounding its reasoning in visual evidence.

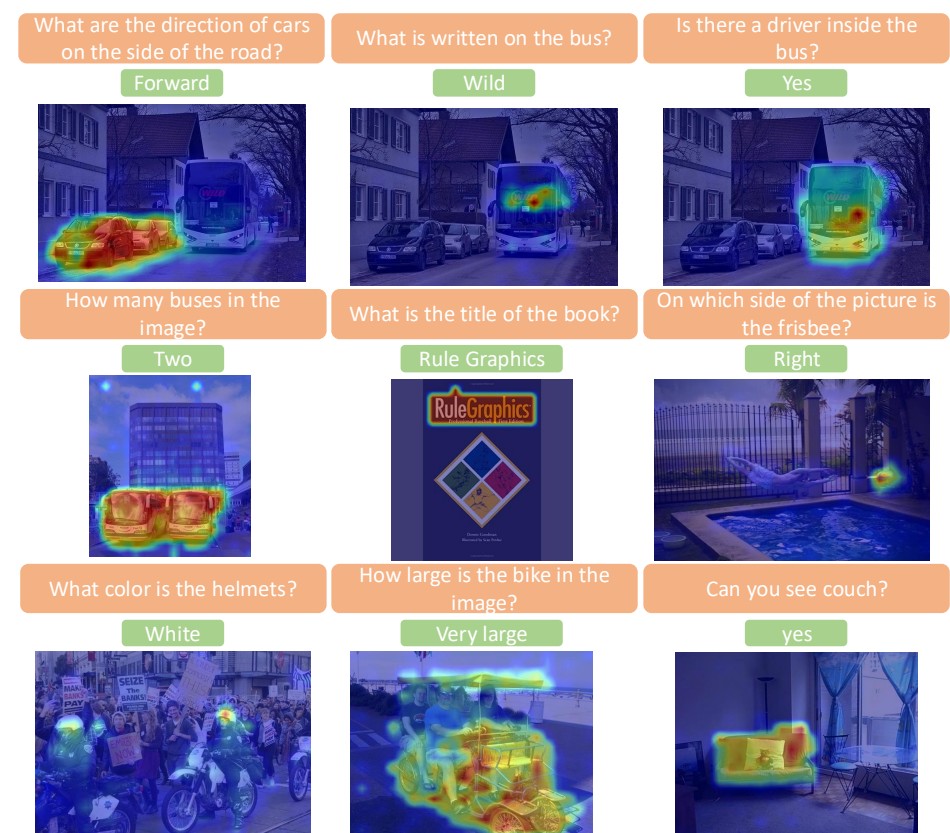

Figure 8: Qualitative examples of **Grounded-based Understanding**. For each question, we show the predicted answer and the corresponding attention map. The highlighted regions align well with the queried visual entities, e.g., focusing on the cars when answering their direction, the bus window when detecting the driver, or the frisbee when asked about its position. These examples illustrate how our loss grounds reasoning in the correct image regions while generating accurate answers.

**LLM as Segmentor.** Figure 9 shows qualitative examples where the LLM, guided by Where-to-Look, directly produces high-quality segmentation masks from its attention maps. Compared to the grounded-based understanding setting, these results are sharper and more precise since the model has been trained on referring datasets such as RefCOCO/+/g. For instance, in the cow example, the attention map clearly isolates the cow with the yellow ear tag while excluding the surrounding fence, demonstrating fine-grained separation of objects from distracting background elements. Similarly, in the human-centric examples, the attention focuses tightly on the described players (e.g., the running player or the one in the green shirt), capturing their contours accurately and ignoring irrelevant individuals. These results highlight that, once trained with referring supervision, the LLM itself can function as a reliable segmentor without relying on heavy external decoders.

**Output Masks with SAM Refinement.** Figure 10 presents qualitative examples of final segmentation masks obtained after integrating our attention guidance with SAM. Here, the attention maps generated by the LLM act as priors (via bounding boxes or keypoints), which SAM then refines into high-quality masks. The results demonstrate strong alignment between the target entities and the refined masks across diverse scenarios, including people, food, sports, and indoor settings. For instance, SAM successfully sharpens object boundaries, separates individuals in crowded scenes, and recovers fine details that raw attention maps may blur. These examples highlight that our method not only enables the LLM to act as a segmentor but also synergizes with external tools like SAM for precise, production-ready segmentation outputs.

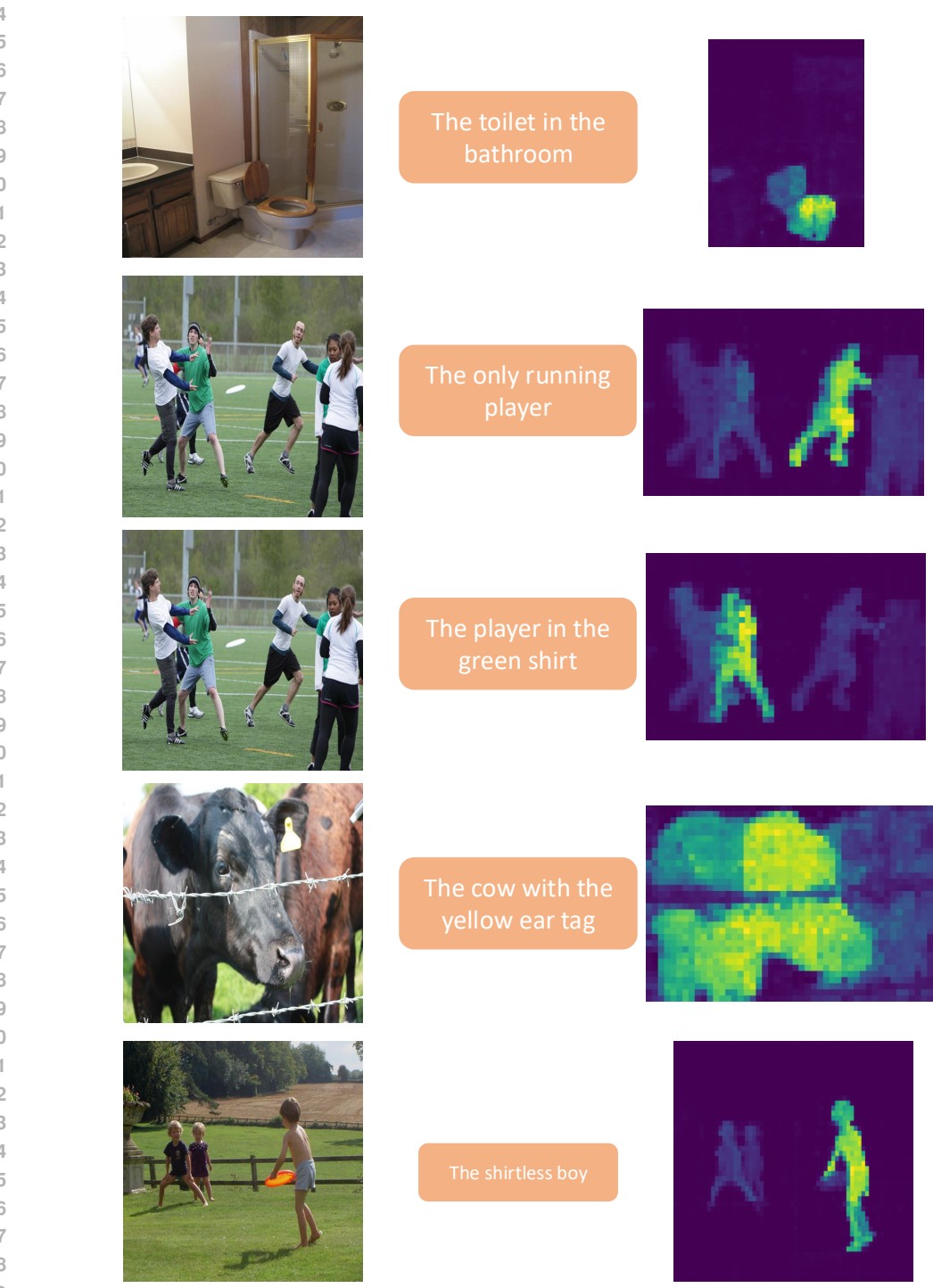

Figure 9: Qualitative examples of **LLM as Segmentor**. Our loss enables the LLM's attention maps to serve directly as segmentation masks. The results show fine-grained localization, e.g., isolating the cow with the yellow ear tag while ignoring the fence, or accurately focusing on specific players (the running player, the one in the green shirt, or the shirtless boy). These examples highlight that the LLM itself can act as a strong segmentor without external decoders.

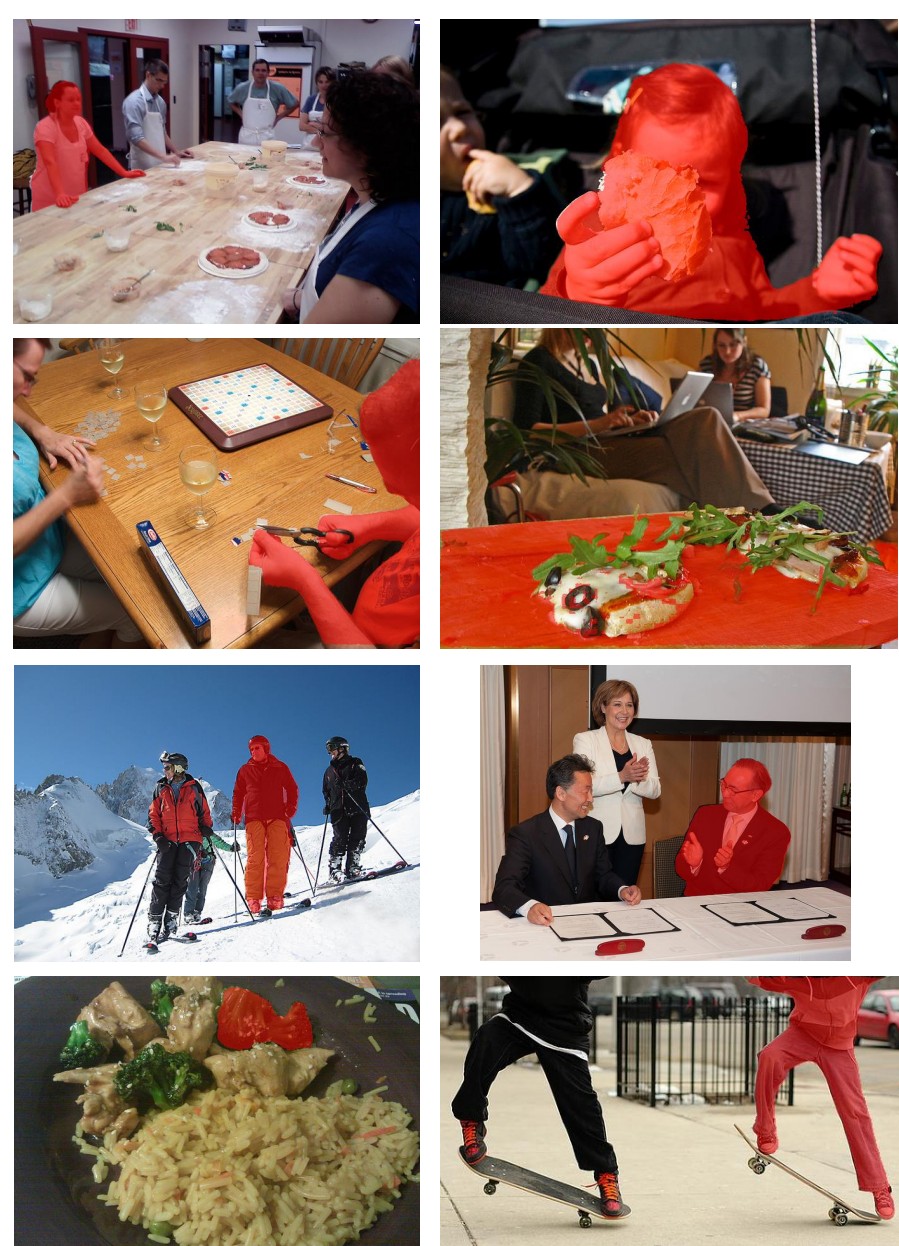

Figure 10: Qualitative examples of **Output Masks with SAM Refinement**. Attention maps from the LLM are used as priors for SAM, which refines them into accurate segmentation masks. Results show robust masks across varied domains, including human activities, food, and complex backgrounds, illustrating the effectiveness of combining our loss with SAM refinement.

## D    USE OF LARGE LANGUAGE MODELS (LLMS)

In accordance with ICLR policy, we disclose our use of LLMs in preparing this paper. LLMs were used exclusively as a general-purpose writing assistant for polishing text, correcting grammar, and improving readability. They were not used for research ideation, experimental design, or interpretation of results, and did not contribute to the scientific content. All ideas, experiments, and conclusions are solely the work of the authors.

