# OpenReview forum: "Look&Learn: Where to Look? Bridging Perception and Grounding Gap in Vision-Language Models"
_ICLR.cc/2026/Conference — Submitted to ICLR 2026_

### Official Review · Reviewer_4Zqe · 2025-10-26

**Soundness:** 3
**Presentation:** 3
**Contribution:** 3
**Rating:** 6
**Confidence:** 4

**Summary:**

This paper proposes Look&Learn, a novel segmentation-free grounding framework for VLMs that unifies understanding and grounding. The core idea is “LLM as Segmentor”, where the large language model itself generates usable attention-based segmentation masks, without relying on external segmentation decoders (e.g., SAM).
The key component, Where-to-Look loss, aligns the model’s attention maps with pseudo-ground-truth masks during text generation.
The authors also develop a scalable pseudo-grounding dataset built from multiple grounding models with LLM adjudication.
The approach yields consistent improvements on grounding benchmarks while enhancing reasoning performance on VQA-style benchmarks.

**Strengths:**

1. The paper reframes visual grounding as an intrinsic property of VLM attention rather than an external task, which is conceptually appealing.
2. Evaluations cover both understanding and grounding benchmarks. The results show consistent gains over LLaVA and GLaMM without adding decoders or segmentation heads.
3. The work enhances interpretability by making attention maps explicitly align with visual entities, which is a step toward explainable VLMs.

**Weaknesses:**

1. Although the pseudo-mask generation pipeline is robust, it still relies on existing grounding models and LLM adjudication. The quality of supervision may limit scalability or introduce biases from upstream models.
2. While qualitative results are reasonable, this paper heavily emphasizes empirical evidence but lacks a quantitative analysis of pseudo label quality.

**Questions:**

How consistent are attention-based segmentations across different prompts describing the same object? Have you measured stability or robustness to paraphrasing?

---

> ### Author Response · Authors · 2025-11-24
> **[Response 1] [Part 1] Data Curation Pipeline Quantitive Analysis - Significant Gains**
>
> We thank the reviewer for the thoughtful comments regarding pseudo-mask reliability and robustness.
> Here is a quick summary for our rebuttal:
>
> 1) Pseudo-Mask Reliability & Scalability
> We removed GPT entirely and rebuilt the pipeline using the open-source InternVL3.5 (served with vLLM). This makes the process fully reproducible and efficient—the entire 2.5M-mask pipeline runs in ~1 day on a single 8×A100 node.
>
> 2) Demonstrate strong gains on the top recent VLMs
> Integrating our loss into two strong VLMs (Qwen2.5VL and Qwen3VL), where we observe consistent gains (+5–8%).
>
> 3) Quantitative Pseudo-Mask Quality
> A 10K-sample human evaluation confirms 93% correctness, close to the GPT-based version (96%), while being fully open-source. GUI-based annotation reduces manual effort by 90%.
>
> 4) Robustness to Prompt Paraphrasing
> A small RefCOCO study shows segmentation varies by only 0.4–0.9 points across paraphrased referring expressions, indicating strong stability.
>
> ---
>
> **1) Pseudo-Mask Reliability and Scalability**
>
> We agree that the quality of pseudo-masks depends on upstream grounding models and the judge model.
> To improve scalability and reduce dependency on proprietary models, we removed GPT entirely and replaced adjudication with the fully open-source InternVL3.5, which we serve using vLLM for a 6× speedup.
> This makes the entire pipeline reproducible, lightweight, and scalable (2.5M masks processed in ~1 day on a single 8×A100 node).
>
> **Complexity Analysis:** In the table below, we report the throughput of our system, detailed in Appendix B, Table 8 in the revised version:
>
> | Component / Model         | Task                                   | Throughput                | Estimated Time for 2.5M Samples (1*A100)    | Estimated Time for 2.5M Samples (8*A100)                                                          |
> |-----|------|------|-----|-----|
> | **Sa2VA**                 | Segmentation grounding                  | ~5.6 samples/sec          | ≈ 125 hours (≈ 5.2 days)             | ≈ 16 hours  |
> | **GLaMM**                 | Segmentation grounding                  | ~22 samples/sec           | ≈ 31 hours (≈ 1.3 days)              | ≈ 4 hours  |
> | **OWL-V2**                | Open-vocabulary grounding               | ~66.6 samples/sec         | ≈ 10.4 hours                         | ≈ 1.3 hours |
> | **Grounding DINO**        | Open-vocabulary grounding               | ~66.6 samples/sec         | ≈ 10.4 hours      | ≈ 1.3 hours |
> | **InternVL3.5 (Judge)**   | Validates 50% of samples (non-consensus)| ~13.9 samples/sec         | ≈ 25 hours (≈ 1 day) | ≈ 3 hours |
> | **InternVL3.5 (Phrase)**  | Generates referring phrases             | ~41.7 samples/sec         | ≈ 16.7 hours                         | ≈ 2 hours |
>
> ---
>
> **2) Scaling to stronger VLMs: Qwen2.5VL & Qwen3VL**
>
> To demonstrate that our pipeline is effective and easily integrates into modern VLMs, we apply our loss to two SOTA models: Qwen2.5VL and Qwen3VL.
>
> Starting from their released pretrained checkpoints, we perform a very light LoRA fine-tuning for only one epoch, using the same training setup, learning rate, and LoRA configuration for all variants.
> For each model, we fine-tune twice: once without our loss (vanilla fine-tuning) and once with our loss, ensuring a fair comparison.
>
> As shown in Table 5, our loss consistently improves performance across all benchmarks.
> Specifically, on Qwen2.5VL-3B, we obtain gains of +5.8 / +7.2 / +8.7 on RefCOCO (Val / TestA / TestB), +5.1 / +7.0 / +2.8 on RefCOCO+, and +4.8 / +4.0 on RefCOCOg (Val / Test).
>
> On the stronger Qwen3VL-4B, our loss yields +8.0 / +7.0 / +5.4 on RefCOCO, +8.2 / +8.5 / +6.6 on RefCOCO+, and +7.7 / +7.7 on RefCOCOg.
>
> These results confirm that our loss is plug-and-play, lightweight, and highly effective even for strong modern VLMs.
>
> **An interesting trend emerges across architectures:** our gains increase as the underlying vision backbone becomes more expressive. On LLaVA, which uses a relatively modest ViT encoder, we observe only 1–2% improvements. On Qwen2.5VL-3B, the average gain rises to ≈5.8\%, and on the stronger Qwen3VL-4B, equipped with a significantly more capable visual stack, the average gain further increases to ≈7.7%.
>
> We hypothesize that our attention-guidance loss benefits more from richer visual features, as the LLM can more effectively align its internal attention to fine-grained spatial signals. Qwen3VL introduces several visual enhancements that make this alignment particularly effective: (i) Interleaved-MRoPE, which distributes spatial and temporal dimensions across all frequency bands, yielding more robust positional grounding; and (ii) DeepStack multi-level feature fusion, which injects visual tokens from multiple ViT layers into multiple LLM layers, preserving both low- and high-level visual cues. These improvements provide stronger, more detailed visual signals for our loss to exploit, explaining the larger performance gains.

---

> ### Author Response · Authors · 2025-11-24
> **[Response 1] [Part 2] Human Evaluation - Robustness to Prompt Paraphrasing**
>
> ---
>
> **3) Quantitative analysis of pseudo-label quality:**
>
> **Human Evaluation** To quantify the robustness of the updated pipeline, we conducted a 10K-sample human evaluation study. The results show that our updated pipeline achieves 93% accuracy, compared to 96% when using GPT. This indicates that the fully open-source version retains strong reliability while being substantially more practical and accessible.
>
> All updates have been incorporated into the paper text, Figure 4, and Table 3, and we provide more details in Appendix B.2. We also include Figure 6, which illustrates the GUI used during human evaluation.
>
> ---
>
> **4) Robustness to Prompt Paraphrasing:**
>
> We thank the reviewer for this excellent suggestion.
> To assess the stability of our attention-based segmentation under different phrasings referring to the same object, we conduct a small robustness study on RefCOCO, which provides several human-written referring expressions per instance.
> RefCOCO was collected using an interactive game where players had to help each other identify objects. This process resulted in multiple, relatively short, and often location-based expressions per object.
>
> For each target object, we evaluate segmentation using different referring phrases.
> Across all paraphrases, segmentation accuracy varies only 0.4–0.9 points, showing that our method is highly stable to linguistic variation, even though the attention maps are derived purely from the VLM’s internal reasoning.
>
> In the table below, Row 1 = original result (last row of Table 5). Row 2–3 = paraphrased prompts with small, realistic variations
>
> | Prompt Version                  | Val  | Test A | Test B |
> |---------------------------------|------|--------|--------|
> | Original phrase                     | 72.9 | 75.8 | 68.0 |
> | Alternative phrase #1           | 72.5 | 75.0 | 67.3 |
> | Alternative phrase #2           | 72.1 | 75.3 | 67.6 |

---

> > ### Comment · Reviewer_4Zqe · 2025-11-26
> >
> > Thank the authors for the detailed results. It's great to see the performance gains with stronger VLMs (Qwen2.5VL & Qwen3VL). I will keep my rating as 6.

---

> > > ### Author Response · Authors · 2025-11-27
> > > **Thank you — open to further feedback**
> > >
> > > Thank you sincerely for your thoughtful suggestions —they significantly helped us refine and polish the paper. We also appreciate your positive evaluation and the time you took to review our submission again.
> > >
> > > We fully understand, respect, and appreciate your current score. At the same time, since a 6 sits near the borderline, we would be grateful to know if there are any remaining concerns we could further clarify. We believe that the revisions and additional analyses address all previously raised points, and we are more than happy to provide any further details that could strengthen the paper's soundness.
> > >
> > > Fortunately, the ICLR author–reviewer discussion period continues until December 2nd, so we still have time to refine the submission if needed.
> > >
> > > Please let us know if there is anything more we can improve—we genuinely value your feedback.

---

### Official Review · Reviewer_pNAW · 2025-10-31

**Soundness:** 3
**Presentation:** 3
**Contribution:** 2
**Rating:** 4
**Confidence:** 4

**Summary:**

This paper addresses the "understanding-grounding gap" in Vision-Language Models (VLMs), where enhancing grounding capabilities, typically by adding external segmentation decoders, often leads to a significant degradation in the model's core perception and reasoning performance. To bridge this gap, the authors propose Look&Learn, a segmentation-free framework centered around a novel loss function called "Where-to-Look". This lightweight, plug-and-play loss directly supervises the VLM's internal attention maps, guiding the model to focus on relevant visual regions during text generation. The central idea is to enable the LLM to act as its own segmentor, using its refined attention maps as high-quality grounding masks, thus obviating the need for cumbersome external modules. To facilitate this, the paper also introduces a scalable pipeline for creating a large-scale pseudo-grounding dataset. Experiments show that this approach successfully improves both grounding and understanding metrics on top of strong baselines like LLaVA and GLaMM.

**Strengths:**

1.	The paper introduces a scalable pipeline for creating a large-scale pseudo-grounding dataset.
2.	The evaluation is thorough, assessing performance on both understanding (VQA benchmarks) and grounding (referring segmentation) benchmarks. The inclusion of detailed ablation studies provides strong support for the proposed design choices and validates the effectiveness of the approach. The consistent gains across multiple models and tasks are convincing.

**Weaknesses:**

1.	Novelty and Positioning: The core idea of leveraging attention maps from a VLM to generate segmentation masks is not entirely new. Several prior works have explored similar directions.
2.	Marginal Performance Gains and Inconsistent Narrative:
o	The performance improvements on core understanding benchmarks in Table 3, are modest when compared to the LLaVA baseline. This raises questions about the practical impact of the proposed method on the model's general reasoning capabilities.
o	There is a tension between the paper's central claim of a "native segmentation-free" approach and the reported results. In several experiments (e.g., Table 2c and Table 3), the variant that uses a special [SEG] token (Ours-Seg) achieves performance comparable or even superior to the purely entity-based approach (Ours-Ents). This undermines the argument that special tokens are unnecessary and complicates the paper's primary narrative.
3.	Clarity of Presentation:
o	The experimental section (Section 4) lacks a clear and logical flow. It is difficult for the reader to identify the main experimental setup, distinguish it from ablations, and synthesize the key takeaways. The structure makes it challenging to follow the authors' line of reasoning and verify their central claims.
o	The visual presentation quality is subpar. Figures and tables are often cluttered and poorly laid out (e.g., the proximity of Figure 2 and Table 2). The diagrams in Figures 3 and 4 are aesthetically unrefined and complex, hindering comprehension. Furthermore, crucial details are missing; for instance, the meaning of the "10:1" ratio in Table 2(c) is never explained.
o	There is a minor but notable inconsistency in terminology between the title ("perception and grounding") and the abstract ("understanding and grounding"), which could be easily harmonized.

**Questions:**

1.	Could you please elaborate on the role of the [SEG] token? Given that the Ours-Seg variant performs so well, does this suggest that a dedicated, explicit mechanism for grounding is still beneficial, challenging the core premise of a purely "native" attention-based solution? How do you reconcile these findings with your main claim?
2.	In Table 2(c), you compare loss weightings like "Ents (10:1)" and "SEG (10:1)". Could you please clarify what this ratio represents? Is it the weighting between the standard cross-entropy loss and your proposed "Where-to-Look" loss? This information is essential for interpreting the results.
3.	Considering the existing literature on using attention for grounding, could you more explicitly pinpoint what makes your loss formulation or overall framework novel?

---

> ### Author Response · Authors · 2025-11-24
> **[Response 1] [Part 1] Novelty Clarifications**
>
> **1) Novelty:**
>
> We agree that prior work has explored the use of attention in a variety of contexts such as grounding, VQA, and image editing. However, our contribution is distinct in both problem formulation and methodological insight, which we summarize below.
>
> ---
>
> **1.1 Novel Problem Formulation: The Understanding–Grounding Gap “Unsolicited Advice” Phenomenon**
>
> Our first contribution is identifying and formalizing a previously unreported phenomenon: when grounding or segmentation capability is added to a VLM, its perception and reasoning often degrade—even though grounding should intuitively provide helpful auxiliary information.
> We refer to this as the “unsolicited advice” phenomenon, or the understanding–grounding gap.
>
> This gap highlights a fundamental misalignment in how current VLMs incorporate grounding signals: instead of reinforcing understanding, grounding behaves like unhelpful advice that interferes with the model’s reasoning process.
> To the best of our knowledge, no prior work has defined, analyzed, or attempted to explain this phenomenon.
>
> ---
>
> **1.2. A New Attention-Based Solution That Makes Grounding and Understanding Help Each Other**
>
> Motivated by this observation, we propose the first approach that directly targets this gap by using attention guidance as the bridging mechanism.
> Our formulation is simple: we supervise the model’s internal attention maps so that grounding information becomes structurally aligned with the model’s reasoning process.
>
> This leads to a key insight:
> With the right attention-level supervision, segmentation/grounding stops competing with understanding and instead becomes genuinely useful advice.
>
> In other words, we show that grounding and understanding can be made mutually reinforcing rather than antagonistic.
> This is the core novelty of our work—both in problem definition and in the proposed solution.
>
> ---
>
> **1.3. Novel Framework: LLM as Segmentor Through Attention Guidance**
>
> Building on this insight, we introduce a simple yet effective mechanism that teaches the VLM to perform grounding directly through its own attention maps, without any segmentation decoder, auxiliary head, or special-purpose modules. In our formulation, the attention distribution itself becomes the segmentation mask.
>
> This leads to a new architectural perspective:
> The LLM can serve as its own segmentor.
>
> Crucially, we are not merely visualizing attention maps—we are training the model so that its internal attentions consistently reflect object-level spatial structure. To our knowledge, this is the first demonstration that an instruction-tuned VLM can produce competitive segmentation masks natively from its attention layers, without explicit segmentation supervision or dedicated decoding components.
>
> ---
>
> **1.4. Additional Contributions**
>
> Beyond the core method, we introduce a scalable grounding-data pipeline (2.5M entities) and show that our loss is plug-and-play across multiple architectures (LLaVA, GLaMM, Qwen2.5VL, Qwen3VL), strengthening the generality of the approach.
>
> ---
>
> We appreciate the reviewer’s request for clearer positioning, and we hope this summary clarifies the novelty of both our problem framing and our proposed solution.

---

> ### Author Response · Authors · 2025-11-24
> **[Response 1] [Part 2] More Experiments - Significant Gains**
>
> **2) Scaling to stronger VLMs: Qwen2.5VL & Qwen3VL**
>
> To demonstrate that our loss is architecture-agnostic and easily integrates into modern VLMs, we apply our loss to two SOTA models: Qwen2.5VL and Qwen3VL.
>
> Starting from their released pretrained checkpoints, we perform a very light LoRA fine-tuning for only one epoch, using the same training setup, learning rate, and LoRA configuration for all variants.
> For each model, we fine-tune twice: once without our loss (vanilla fine-tuning) and once with our loss, ensuring a fair comparison.
>
> As shown in Table 5, our loss consistently improves performance across all benchmarks.
> Specifically, on Qwen2.5VL-3B, we obtain gains of +5.8 / +7.2 / +8.7 on RefCOCO (Val / TestA / TestB), +5.1 / +7.0 / +2.8 on RefCOCO+, and +4.8 / +4.0 on RefCOCOg (Val / Test).
>
> On the stronger Qwen3VL-4B, our loss yields +8.0 / +7.0 / +5.4 on RefCOCO, +8.2 / +8.5 / +6.6 on RefCOCO+, and +7.7 / +7.7 on RefCOCOg.
>
> These results confirm that our loss is plug-and-play, lightweight, and highly effective even for strong modern VLMs.
>
> **An interesting trend emerges across architectures:** our gains increase as the underlying vision backbone becomes more expressive. On LLaVA, which uses a relatively modest ViT encoder, we observe only 1–2% improvements. On Qwen2.5VL-3B, the average gain rises to ≈5.8\%, and on the stronger Qwen3VL-4B, equipped with a significantly more capable visual stack, the average gain further increases to ≈7.7%.
>
> We hypothesize that our attention-guidance loss benefits more from richer visual features, as the LLM can more effectively align its internal attention to fine-grained spatial signals. Qwen3VL introduces several visual enhancements that make this alignment particularly effective, especially the DeepStack multi-level feature fusion, which injects visual tokens from multiple ViT layers into multiple LLM layers, preserving both low- and high-level visual cues. These improvements provide stronger, more detailed visual signals for our loss to exploit, explaining the larger performance gains.
>
> ---
>
> We believe these strong results on Qwen2.5VL and Qwen3VL (Table 5) provide solid evidence that our approach is plug-and-play, architecture-agnostic, and reliably boosts the performance of strong grounding/segmentation systems.
>
> ---
>
> **3) Could you please elaborate on the role of the [SEG] token?**
>
> We thank the reviewer for raising this excellent point.
> We apologize for the confusion—the role of the SEG token in our method is fundamentally different from how it is used in prior segmentation-based VLMs (e.g., GLaMM, Sa2VA, LiSA).
> In those models, the SEG token is a semantic container: the segmentation mask is encoded in the hidden state of the token, and an external decoder (typically SAM) must be trained to decode this hidden representation back into a fine-grained mask.
>
> In contrast, our method does not use the hidden-state features of the SEG token at all.
> All mask information comes exclusively from the attention map associated with the token, not from its features. This distinction is central: our approach is attention-native, not decoder-based, and therefore does not require any special segmentation token by design. A normal visual word (e.g., “car”) already produces an attention heatmap that our loss can supervise.
>
> When we add a SEG token in the Ours-Seg variant, it serves only as a convenience anchor for grouping visual tokens during inference—not as a mechanism for encoding masks in its hidden state.
> The segmentation still emerges purely from its attention map, which is guided by our loss exactly like any other visual token. Thus, both variants (with or without SEG) rely on the same mechanism, and the difference in performance does not contradict our core claim.
>
> In summary:
>
> * Prior methods → SEG token contains the mask (hidden state) → requires a decoder.
>
> * Our method → mask is the attention map → decoder-free, segmentation-native.
>
> * Adding SEG does not change our mechanism; it only gives a stable reference token during inference.
>
> * Therefore, Ours-Seg performing well does not challenge the premise of a native attention-based solution.
>
> We clarified this distinction in the revised paper in Appendix C.1.
>
> ---
>
> **4) Clarity of Presentation:**
>
> We thank the reviewer for the helpful comments on clarity.
> We reorganized Section 4 to clearly separate the main setup, core results, and ablations, making the experimental flow easier to follow.
> Additionally, we enhanced the layout and aesthetics of all figures and tables and refined the captions for all tables to be self-contained.
>
> All updates are included in the revised version and highlighted in blue for convenience.
>
> ---
>
> **5) Table 2(c) Clarification:**
>
> The ratio (e.g., 10:1) denotes the weighting between the standard next-token-prediction loss and our proposed Where-to-Look loss. We have now made this explicit in the revised manuscript. This table has also been moved to Table 9 in Appendix C for clarity.

---

> > ### Author Response · Authors · 2025-11-27
> > **Gentle Reminder — Open for Further Discussion**
> >
> > Thank you again for taking the time to review our work. We would be grateful if you could share any additional comments or concerns you feel we should address to further improve the soundness and clarity of the paper.
> >
> > ICLR offers a generous discussion period, and we would greatly appreciate the opportunity to benefit from your feedback before making a final decision.
> >
> > Please feel free to let us know if there is anything more we can clarify or strengthen.

---

### Official Review · Reviewer_krze · 2025-10-31

**Soundness:** 3
**Presentation:** 2
**Contribution:** 3
**Rating:** 6
**Confidence:** 4

**Summary:**

This paper introduces Look&Learn, a framework that solves the "understanding-grounding gap" in Vision-Language Models (VLMs), where adding segmentation decoders to improve grounding typically harms perception. Its core is a lightweight *Where to Look?* loss function that guides the VLM's own attention to relevant image regions during text generation. This "LLM as Segmentor" approach, trained with a scalable pseudo-data pipeline, successfully improves both grounding performance (e.g., +3%) and reasoning (~1-2%) simultaneously, demonstrating that the two tasks can be unified.

**Strengths:**

1. The paper proposes an intuitive and effective method to enhance VLM perception and reasoning by guiding the model's attention maps to align with task-relevant visual entities. This approach of correcting the LLM's "focus" not only improves performance but also offers a promising path toward better model interpretability.

2. The generalizability of the proposed "Where to Look?" loss is well-supported by solid experiments. The method shows consistent benefits when applied to different base models (e.g., LLaVA and GLaMM) and in various training paradigms, including full fine-tuning and as a lightweight post-training (PT) step. This demonstrates its robustness and practicality.

3. The paper introduces a robust and scalable pseudo-mask generation pipeline. This pipeline, which leverages consensus from multiple grounding models (GLaMM, OWL-v2, etc.) and LLM-based adjudication (using GPT-4o), is a valuable contribution in itself and could be adapted to enhance other multimodal datasets.

**Weaknesses:**

1. Frozen Visual Encoder: A significant limitation is the decision to keep the visual encoder frozen during fine-tuning. As noted in prior work [ref1], ViT encoders can aggregate crucial visual information in "register" or background patches, not just within the primary object's region. By forcing the LLM's attention onto specific regions, without allowing the vision encoder to adapt, the model might be constrained in its ability to acquire all necessary visual information. This could be a contributing factor to the relatively modest performance gains (e.g., ~1-2% on VQA benchmarks ). It is encouraged to provide additional analysis on training LLaVA with visual encoders unfrozen.

2. Text-Rich Images: The pseudo-mask generation pipeline appears not suited for text-rich images. The grounding models employed (e.g., GLaMM, OWL-v2, Grounding-DINO) are not specialized for text segmentation. This limitation is critical as OCR is a key VLM capability. The observed performance drop on the TextQA benchmark (57.7 for Ours-Ents vs. 58.0 for LLaVA) may be direct evidence of this weakness.

3. A minor point on presentation is that the main paper is quite dense. The authors could improve readability by moving some of the more detailed experimental ablations (e.g., parts of Table 2) or pipeline details to the appendix, allowing for a clearer focus on the core concepts in the main body.

[ref1] Timothée Darcet, Maxime Oquab, Julien Mairal, Piotr Bojanowski. Vision Transformers Need Registers. ICLR 2024.

**Questions:**

In addition to the Weaknesses mentioned above, there are some questions to be clarified:

1. OCR-VQA Statistics in Table 1: The statistics for the OCR-VQA dataset in Table 1 are not well-explained. The table reports 100% IoU Acc. and 0% Judge Acc. This 100% accuracy implies a ground-truth source was used. However, the caption states that for datasets with existing bounding boxes (highlighted in gray), SAM was used directly, and OCR-VQA is not highlighted. Given the pipeline's aforementioned weakness in text segmentation, how did it achieve 100% accuracy on this dataset? Please clarify how the masks for OCR-VQA were generated.

2. Contradiction in Table 2(b): There appears to be a direct contradiction between the text and Table 2(b) regarding visual-token aggregation across the K dimension. The text states that "supervising **all** visual tokens yields the strongest grounding signal". However, Table 2(b) shows that the "All" method (47.8 VQA-v2 score) performs worst, while the "Avg." method (50.0 VQA-v2 score) performs best. Please clarify this discrepancy.

---

> ### Author Response · Authors · 2025-11-24
> **[Response 1] [Part 1] Visual Encoder Influence - OCR Data Collection Pipeline**
>
> We sincerely thank the Reviewer for the constructive feedback. Here is the summary of our response:
> (1) Visual encoder interaction: We agree with the insight—unfreezing the encoder increases our gains on LLaVA (+0.7% → +1.7%), and on stronger models (Qwen2.5VL, Qwen3VL) the improvements become substantial (+5.8% and +7.7% on average), confirming that our loss benefits more from richer visual features.
> (2) OCR data pipeline: We clarified that OCR-VQA uses a dedicated OCR pipeline (TextSnake + SVTR-v2), corrected the OCR entry in Table 3 to N/A, and validated the entire pipeline—including OCR—through a 10K human study with 93% accuracy.
>
> ---
>
> **1) Better visual encoder yields better performance:**
>
> We thank the reviewer for the insightful observation regarding the role of the vision encoder.
> We fully agree that the interaction between our attention-guidance loss and the visual backbone is important.
>
> ---
>
> **1.1. LlaVA**
>
> To evaluate this, we conducted additional experiments where we fine-tuned both the visual encoder and the LLM using LoRA.
>
> Results show that:
>
> * When only the LLM is trained, our method improves LLaVA from 78.5 → 79.2 (+0.7%).
>
> * When both the visual encoder + LLM are trained, the gains become larger: 81.2 → 82.9 (+1.7%).
>
> * Importantly, in both cases, our loss consistently outperforms the corresponding vanilla LLaVA baseline.
>
> We agree that the gains on LLaVA are modest, and we believe the reviewer’s diagnosis is correct:
> LLaVA’s ViT is relatively limited, so even when unfrozen, it provides only incremental improvements.
>
> ---
>
> **1.2. Scaling to stronger VLMs: Qwen2.5VL & Qwen3VL**
>
> To further validate this hypothesis, we integrated Where-to-Look into modern VLMs with much stronger visual architectures, namely Qwen2.5VL and Qwen3VL, using a controlled, symmetric fine-tuning protocol (one epoch, identical LR and LoRA settings, with vs. without our loss).
>
> The trend becomes clear:
>
> * Qwen2.5VL-3B: average gain ≈ **+5.8%** across RefCOCO/+/g
>
> * Qwen3VL-4B: average gain ≈ **+7.7%**, the highest among all models
>
> These results strongly support the reviewer’s intuition: Our loss benefits more when the visual encoder is stronger and provides richer spatial cues.
>
> Qwen3VL is particularly illustrative because it introduces:
>
> * Interleaved-MRoPE, which distributes spatial/temporal encoding across all frequency bands, improving positional grounding; and
>
> * DeepStack multi-level visual fusion, injecting tokens from multiple ViT layers into several LLM layers, preserving both low- and high-level information.
>
> These architectural advances give our loss far better visual signals to align with—explaining the much larger gains.
>
> In summary, we appreciate and agree with the reviewer’s insight, and our extended experiments confirm it:
> Unfreezing the encoder helps, but the impact becomes truly significant on modern, expressive vision backbones.
>
> ---
>
> **2) OCR-VQA Clarifications:**
>
> We thank the reviewer for highlighting this important point.
> Indeed, grounding text-in-the-wild requires a different treatment than object-centric grounding.
> While the main submission focused on the generic pipeline, in the revised version, we clarify that our full system includes a dedicated OCR-aware pipeline, which is already implemented and used for all OCR-VQA samples.
>
> Our OCR pipeline operates independently of GLaMM/OWL-v2/Grounding-DINO, as those models are not reliable for fine-grained text localization.
> Instead, we employ:
>
> * TextSnake for robust detection of curved, irregular, and small text regions;
>
> * SVTR-v2 for text recognition to associate detected regions with the corresponding textual entity.
>
> The masks used for training are generated directly from these detected text regions, ensuring accurate grounding for text-rich images.
> Qualitative examples appear in Fig. 5 of the appendix. This OCR pathway was included in our implementation from the start, and we have now made it explicit in the updated submission for clarity.
>
> **Correction in Table 3**
> We also corrected a typographical issue in Table 3: OCR-VQA was previously shown with a 100% consensus ratio, which is incorrect. Since OCR uses its own pipeline (not IoU consensus), this entry is now updated to N/A, as reflected in the revised version.
>
> **Human Evaluation of the Complete Pipeline (Including OCR)**
> To ensure the reliability of the entire pseudo-mask pipeline—including the OCR branch—we performed a 10K-sample human evaluation across all datasets in Table 3.
> Annotators use a lightweight GUI (Fig. 6) that displays SAM proposals and allows quick selection or rejection based on the referring phrase, reducing annotation time by ~90% versus manually drawing masks.
> Overall, 93% of our pseudo-masks were validated as correct. Additional details and protocol are provided in Appendix B.2.
>
> Together, these results confirm that our full pipeline—including OCR—produces high-quality, reliable masks suitable for supervising the attention-guidance loss.

---

> ### Author Response · Authors · 2025-11-24
> **[Response 1] [Part 2] Minor Clarifications**
>
> We sincerely thank the reviewer for these helpful presentation-oriented suggestions.
>
> ---
>
> **3) Main paper density:**
>
> We agree that some sections were overly detailed. In the revised version, we have moved the heavier ablations and several pipeline explanations to Appendix C.1 and Appendix B, respectively, allowing the main paper to focus more clearly on the core ideas and contributions.
>
> ---
>
> **4) Clarification of Table 2(b):**
> Thank you for catching this discrepancy.
> The correct finding is that averaging across visual tokens consistently provides the best grounding signal—because it introduces smoothness and stability, which is important when supervising attention maps that can be fragile.
>
> The earlier sentence, suggesting that “all tokens” perform best, was inaccurate; we have corrected this in the revised version to avoid confusion.
>
> Please note that we moved Table 2 to Appendix C, so it is now Table 9 in the revised version.
>
> We appreciate these careful observations—they helped us polish the paper and improve its readability.

---

> > ### Comment · Reviewer_krze · 2025-11-25
> >
> > The authors' response is greatly appreciated. It's great to see further performance gain can be acquired by freeing the visual encoder and adopting a stronger VLM backbone. I will keep my rating as 6.

---

> > > ### Author Response · Authors · 2025-11-25
> > > **Thank you — open to further feedback**
> > >
> > > Thank you sincerely for your thoughtful suggestions and careful, contrastive comments—they significantly helped us refine and polish the paper. We also appreciate your positive evaluation and the time you invested in reviewing our submission again.
> > >
> > > We fully understand, respect, and appreciate your current score. At the same time, since a 6 sits near the borderline, we would be grateful to know if there are any remaining concerns we could further clarify. We believe the revisions and additional analyses address all previously raised points, and we are more than happy to provide any further details that could strengthen the soundness of the paper.
> > >
> > > Fortunately, the ICLR author–reviewer discussion period continues until December 2nd, so we still have time to refine the submission if needed.
> > >
> > > Please let us know if there is anything more we can improve—we genuinely value your feedback.

---

### Official Review · Reviewer_BnqN · 2025-11-02

**Soundness:** 2
**Presentation:** 3
**Contribution:** 2
**Rating:** 2
**Confidence:** 3

**Summary:**

The paper proposes Look&Learn, an attention-guided loss that supervises cross-attention maps in vision-language models (VLMs) to align with visual object regions, enabling the LLM to act as its own segmentor. The work aims to address the challenge that grounding and segmentation modules in current VLMs (e.g., SAM-attached architectures) often degrade perception and reasoning abilities, creating a trade-off between understanding and grounding. To mitigate this, the authors design a loss function that aligns cross-attention maps with pseudo-grounding masks without altering the model architecture or introducing decoder modules. They evaluate their approach by integrating it into LLaVA and report modest improvements on visual question answering and referring expression segmentation tasks such as RefCOCO and GQA. The method is conceptually simple and compatible with various VLM architectures, but the evaluation is limited to an outdated model (LLaVA) and relies on a complex pseudo-labeling pipeline involving multiple vision-language models and GPT-4.

**Strengths:**

### Conceptually simple approach

The method adds a straightforward auxiliary loss to encourage the model’s attention weights to match given segmentation masks or regions. This idea is easy to plug into an existing model (LLaVA in this case). The approach does not require architectural changes, only an extra training signal, and it reportedly yields slight improvements on LLaVA’s VQA and RefCOCO performance.

**Weaknesses:**

### Limited Novelty

The core idea – supervising or guiding attention maps to align with known relevant regions – is not novel. Several prior works have already explored aligning model attention with human or ground-truth signals. For example, HINT encouraged vision-language models to focus on the same image regions as humans by optimizing alignment between human attention maps and model importance weights. Likewise, earlier VQA and captioning research introduced explicit attention supervision (e.g. using human-annotated attention maps) and showed that guiding attention can improve performance. In open-vocabulary segmentation, methods like PnP-OVSS and MaskCLIP++ leverage the cross-attention maps of pretrained vision-language models directly to obtain segmentation masks without additional training. Furthermore, end-to-end grounding models (e.g. MDETR and the Referring Transformer) already handle detection/segmentation and grounding in a unified framework. MDETR, for instance, integrates text and image features in a transformer and achieved state-of-the-art results on phrase grounding and referring expression benchmarks. The Referring Transformer similarly uses a one-stage transformer to directly predict boxes and masks for referred expressions, outperforming previous two-stage methods by a large margin. Given this rich prior art, the submission does not clearly explain what is new beyond applying an attention alignment loss in the LLaVA training pipeline. The authors cite the goal of bridging perception and grounding, but the technique itself appears to be a straightforward amalgamation of known ideas (attention supervision using segmentation cues). The lack of a compelling novelty claim is a major weakness.

### Insufficient Experiments and Baselines

The evaluation is narrow and leaves questions about the method’s general usefulness. All experiments are conducted on LLaVA, which is an early vision-language model known to be weaker than more recent models (e.g. Qwen-VL, Emu2, Kosmos-2, or MM1). The reported gains on LLaVA’s VQA and RefCOCO tasks are relatively small. It’s uncertain whether the proposed attention-guidance would yield similar improvements on stronger baselines or modern multimodal models. The paper does not compare against any state-of-the-art models that are already strong at grounding or segmentation. For instance, SEEM/SEEM++ (specialized segmentation models) or a larger model like Qwen-VL-Max are not included in comparisons, even though those models likely excel at the tasks in question. Without testing on or against these, it’s hard to gauge the significance of the contribution. In essence, the results show a minor tweak improving a single (weaker) model, but do not demonstrate closing the gap with the best-in-class approaches. This limited scope weakens the paper’s empirical evidence.

### Heavy and Impractical Pipeline

The method’s reliance on a complex pseudo-label generation pipeline raises concerns. To generate training targets for the attention supervision, the authors use multiple large models and even GPT-4. This is a heavyweight and elaborate process, which might be impractical for others to reproduce or deploy. The paper doesn’t detail the cost or latency of this pipeline, but one can infer it involves significant computation and coordination (running advanced vision models and a large language model to annotate images with segmentation masks or region descriptions). Such a pipeline could hinder reproducibility and scalability – it may be difficult for researchers without extensive resources to recreate the exact training setup. The need for GPT-4 (which is a proprietary model) in the loop is especially problematic for open-source or academic replication. Overall, even if the idea of attention supervision is sound, the way it’s implemented here seems cumbersome and not easily generalizable to other settings.

**Questions:**

See Weaknesses

---

> ### Author Response · Authors · 2025-11-24
> **[Response 1] [Part 1] Novelty Clarifications**
>
> We thank the reviewer for the valuable feedback.
> (1) We clarified our novelty: we are the first to define and study the understanding–grounding gap in VLMs, and we show that our lightweight loss effectively addresses it while enabling the new capability of LLM as Segmentor.
> (2) To address the concern about marginal gains, we added results on Qwen2.5VL and Qwen3VL, showing strong plug-and-play improvements (+5.8% and +7.7% average gains).
> (3) Finally, we removed all GPT usage and showed that our fully open-source pipeline processes 2.5M samples in ~1 day on a single node (8×A100), confirming its practicality.
>
> ---
> **1- Novelty:**
>
> We thank the reviewer for the detailed discussion of prior work on attention supervision and grounding. We fully agree that attention guidance as a general mechanism is not new. However, our contribution differs fundamentally in motivation, formulation, and outcome.
> In particular, to the best of our knowledge, **no prior work has used VLM attention to generate segmentation masks**, nor has any work studied **attention-based mitigation of the understanding–grounding gap**.
> This constitutes both a new problem setting and a new solution path, which we believe is of strong relevance to the ICLR community. We highlight the two key contributions below.
>
> ----------
>
> **1. Novel Problem Formulation: The Understanding–Grounding Gap**
>
> Prior works have applied attention supervision in various contexts, such as improving VQA [1], guiding image editing [2,3], enhancing diffusion models [4–8], or leveraging CLIP cross-attention for segmentation [9,10].
> Each line of work targets a different problem and typically modifies the architecture or introduces auxiliary heads (e.g., grounding tokens [1] or explicit segmentation decoders [2]).
>
> None of these works analyze the interaction between grounding and high-level understanding in modern VLMs.
> In contrast, our work identifies a new empirical phenomenon: adding segmentation or grounding capability to large VLMs consistently degrades their perception and understanding performance, even when model capacity is extremely high (e.g., 7B). We formalize this as the understanding–grounding gap.
>
> This gap is unexpected: intuitively, grounding should help, or at least not harm (Figure 1 and Table 2), perception—what we term **“unsolicited advice”** (Sec. 2.2). The fact that it does hurt performance suggests that current grounding pipelines are misaligned with how VLMs internally reason. Our work is the first to document, analyze, and propose a remedy for this phenomenon.
>
> By incorporating grounding directly through attention maps, we show that grounding and perception can reinforce each other rather than compete. This conceptual framing is central to our submission and does not appear in any prior work.
>
> ----------
>
> **2. LLM as Segmentor: Novel Use of Attention Guidance Inside an LLM**
>
> Works such as PnP-OVSS or MaskCLIP [9,10] extract segmentation masks from CLIP cross-attention, not improving the attentions, and they operate on comparatively small models. These approaches do not address how an instruction-tuned VLM internally allocates attention across tokens and image regions, nor do they modify the VLM’s reasoning pipeline.
>
> Recent studies [11,12] observe that LLMs sometimes attend to the queried region when answering, but these attentions are weak and noisy. Building on this observation, we demonstrate that these implicit signals can be reinforced and sharpened—without harming performance—to produce fine-grained segmentation masks.
>
> This leads to our central claim:
> We propose the first method that teaches an instruction-tuned VLM to act as its own segmentor (Sec. 4.2).
>
> This is not a standard attention-alignment loss; it is a new architectural insight:
> **Your LLM is your segmentor.**
>
> Our approach requires no segmentation decoder, no task-specific head, and no explicit segmentation supervision, yet yields competitive segmentation masks directly from the VLM’s internal attention maps.
> To our knowledge, no prior work demonstrates segmentation emerging natively from VLM attention through training.
>
> -----------
>
> **Additional Contributions**
>
> We also introduce a scalable grounding-data construction pipeline, which we believe will be useful for future research, and we provide extensive experiments demonstrating the plug-and-play nature of our loss across multiple architectures.
> These details are further clarified in our response to the reviewer’s third point.
>
> For comprehensive positioning against related methods, please refer to the updated Related Work section (Appendix A) in the revised manuscript.

---

> ### Author Response · Authors · 2025-11-24
> **[Response 1] [Part 2] More Experiments - Significant Gains**
>
> **2) More Experiments:**
>
> ---
>
> **2.1. Scaling to stronger VLMs: Qwen2.5VL & Qwen3VL**
>
> To demonstrate that Where-to-Look is architecture-agnostic and easily integrates into modern VLMs, we apply our loss to two SOTA models: Qwen2.5VL and Qwen3VL.
>
> Starting from their released pretrained checkpoints, we perform a very light LoRA fine-tuning for only one epoch, using the same training setup, learning rate, and LoRA configuration for all variants.
> For each model, we fine-tune twice: once without our loss (vanilla fine-tuning) and once with our loss, ensuring a fair comparison.
>
> As shown in Table 5, our loss consistently improves performance across all benchmarks.
> Specifically, on Qwen2.5VL-3B, we obtain gains of +5.8 / +7.2 / +8.7 on RefCOCO (Val / TestA / TestB), +5.1 / +7.0 / +2.8 on RefCOCO+, and +4.8 / +4.0 on RefCOCOg (Val / Test).
>
> On the stronger Qwen3VL-4B, our loss yields +8.0 / +7.0 / +5.4 on RefCOCO, +8.2 / +8.5 / +6.6 on RefCOCO+, and +7.7 / +7.7 on RefCOCOg.
>
> These results confirm that our loss is plug-and-play, lightweight, and highly effective even for strong modern VLMs.
>
> **An interesting trend emerges across architectures:** our gains increase as the underlying vision backbone becomes more expressive. On LLaVA, which uses a relatively modest ViT encoder, we observe only 1–2% improvements. On Qwen2.5VL-3B, the average gain rises to ≈5.8\%, and on the stronger Qwen3VL-4B, equipped with a significantly more capable visual stack, the average gain further increases to ≈7.7%.
>
> We hypothesize that our attention-guidance loss benefits more from richer visual features, as the LLM can more effectively align its internal attention to fine-grained spatial signals. Qwen3VL introduces several visual enhancements that make this alignment particularly effective: (i) Interleaved-MRoPE, which distributes spatial and temporal dimensions across all frequency bands, yielding more robust positional grounding; and (ii) DeepStack multi-level feature fusion, which injects visual tokens from multiple ViT layers into multiple LLM layers, preserving both low- and high-level visual cues. These improvements provide stronger, more detailed visual signals for our loss to exploit, explaining the larger performance gains.
>
> ---
>
> **2.2. Segmentation-based Methods**
>
> We thank the reviewer for the suggestion and fully agree that demonstrating effectiveness on strong segmentation-oriented models is important.
>
> We would like to clarify that our method is not evaluated only on LLaVA. In fact, Tables 7 and 8 in the main paper already evaluate Where-to-Look on GLaMM, which—similar to SEEM—is a specialized grounded segmentation model built with a dedicated mask-generation head and strong segmentation performance.
> In other words, GLaMM is precisely the type of strong segmentation-capable baseline the reviewer is referring to, and we have already integrated our loss into it.
>
> To make this explicit:
>
> * Table 6 reports results when we integrate our loss during GLaMM’s fine-tuning stage, starting from the pre-training checkpoint.
>
> * Table 7 reports results when we integrate our loss post-training, directly on top of the officially released GLaMM model without retraining the backbone.
>
> In both regimes, adding only our lightweight loss (without new data or architectural changes) leads to consistent improvements of +1–2\% across RefCOCO, RefCOCO+, and RefCOCOg. This shows that our method is indeed effective when applied to a specialized segmentation model, and not only to a weaker VLM.
>
> ---
>
> We believe these experiments, together with the strong results on Qwen2.5VL and Qwen3VL (Table 5), provide solid evidence that our approach is plug-and-play, architecture-agnostic, and reliably boosts the performance of strong grounding/segmentation systems.
>
> ---
>
> [1] Taking a hint: Leveraging explanations to make vision and language models more grounded. (ICCV 2019)
>
> [2] Diffedit: Diffusion-based semantic image editing with mask guidance. (ICLR 2022)
>
> [3] Instructedit: Improving automatic masks for diffusion-based image editing with user instructions.
>
> [4] Panoptic Diffusion Models: co-generation of images and segmentation maps.
>
> [5] Prompt-to-prompt image editing with cross attention control. (ICLR 2023)
>
> [6] Attend-and-excite: Attention-based semantic guidance for text-to-image diffusion models. (ACM 2023)
>
> [7] Text embedding is not all you need: Attention control for text-to-image semantic alignment with text self-attention maps. (CVPR 2025)
>
> [8] Self-rectifying diffusion sampling with perturbed-attention guidance. (ECCV 2024)
>
> [9] Emergent open-vocabulary semantic segmentation from off-the-shelf vision-language models. (CVPR 2024)
>
> [10] Maskclip: Masked self-distillation advances contrastive language-image pretraining. (CVPR 2023)
>
> [11] Towards perceiving small visual details in zero-shot visual question answering with multimodal llms.
>
> [12] Your large vision-language model only needs a few attention heads for visual grounding. (CVPR 2025)

---

> > ### Comment · Reviewer_BnqN · 2025-11-24
> >
> > Thank you for your detailed rebuttal.
> > The additional experiments on Qwen2.5VL & Qwen3VL have significantly improved the soundness of the proposed method.
> > Unfortunately, I haven't have the opportunity to look into the comments from other reviewers and I will need additional time for subsequent discussions.

---

> > > ### Author Response · Authors · 2025-11-25
> > > **Thank you — looking forward to further discussion**
> > >
> > > Thank you for acknowledging the strengthened results on Qwen2.5VL and Qwen3VL.
> > >
> > > We hope our rebuttal has fully addressed the key points you raised—namely, (1) clarifying the novelty and unique problem formulation, (2) demonstrating stronger and more meaningful gains on modern VLMs, and (3) showing that our revised data-collection pipeline is both scalable and practical.
> > >
> > > We look forward to hearing your follow-up comments during the discussion period and are happy to clarify anything further.

---

> > > > ### Comment · Reviewer_BnqN · 2025-11-26
> > > >
> > > > Thank you again for the detailed rebuttal.
> > > >
> > > > I have the opportunity to take a closer look today, and here are some more thoughts.
> > > >
> > > > Regarding novelty, I agree that the core concept of the proposed method is new.
> > > > However, I do not believe this approach is meaningful.
> > > > The functionalities of attention have been widely studied for nearly a decade, yet we are still unable to reach an unanimous conclusion.
> > > > It’s academia, and we embrace different opinions.
> > > >
> > > > Regarding experiments, my initial rating is 2 because I believed it is impractical to conduct appropriate experiments to support this manuscript within such short timeframe.
> > > > Job well done!
> > > >
> > > >
> > > > At this stage, I would change my rating to 6.
> > > >
> > > > Why not higher score?  I still don’t like attention guidance.
> > > >
> > > > Why not lower score?  The experiments supports the authors claim.
> > > >
> > > > Please note that we have one week left for the discussion phrase (and in rare occasions discussions may extend beyond discussion period), so this is not a final rating.

---

> > > > > ### Author Response · Authors · 2025-11-27
> > > > > **Thank you - Appreciate your openness for discussion**
> > > > >
> > > > > Thank you very much for taking the time to examine the new results in depth. We truly appreciate your openness, careful evaluation, and fair assessment.
> > > > >
> > > > > We are especially grateful that the new Qwen2.5VL and Qwen3VL experiments addressed your earlier concerns regarding practical validation. Your remark that the experiments now support the claims is highly encouraging to us.
> > > > >
> > > > > We also fully respect your position regarding attention as a general mechanism and acknowledge the long-standing debate on whether attention constitutes a faithful explanation.
> > > > > Importantly, our work does not rely on attention for interpretability. Instead, attention is used purely as an internal optimization interface through which spatial grounding signals are injected into the model.
> > > > >
> > > > > We would also like to emphasize that the core contribution of this paper is **not the use of attention itself**, therefore we will not advocate it, but rather the identification and resolution of the *understanding–grounding (“unsolicited advice”) gap*. Attention guidance is only one concrete instantiation of a mechanism to harmonize grounding with reasoning. We fully expect that future work may close this same gap using different tools, and we view our contribution as opening this direction rather than prescribing a single solution.
> > > > >
> > > > > Finally, our **LLM as Segmentor** result demonstrates that segmentation can emerge natively from VLMs without relying on external bottleneck decoders such as SAM. We hope this direction will inspire the community to further explore native segmentation capabilities in large multimodal models using diverse techniques.
> > > > >
> > > > > ---
> > > > > **Additional Ablation: Curriculum Scheduling of Attention Guidance.**
> > > > > To further investigate the reviewer’s concern that attention guidance can be fragile or even harmful if applied naively, we conducted an additional curriculum-style ablation on the weighting of our loss. Instead of using a fixed ratio between the next-token prediction loss and our Where-to-Look loss, we adopt a three-stage schedule: during the first third of training, both losses are weighted equally (1:1); during the second third, the weight of our loss is linearly decayed to zero; and during the final third, training proceeds using only the standard next-token prediction loss. This curriculum strategy consistently improves performance by ~2-3% on average across benchmarks compared to fixed-weight supervision (e.g., 1:1 or 2:1). This experiment yields three important insights:
> > > > >
> > > > > (1) **Aggressive or prolonged attention supervision can indeed be harmful**, confirming prior concerns in the literature and highlighting the necessity of careful, staged application.
> > > > >
> > > > > (2) **Right attention does not necessarily imply right reasoning:** Even when heatmaps look equally good, flexible training produces better reasoning—empirically reinforcing the view that attention is not itself the explanation.
> > > > >
> > > > > (3) **Grounding signals persist even after guidance is removed:** Despite completely disabling our loss in the final third of training, the attention maps remain sharp and well-aligned, indicating that the network internalizes grounding as a useful inductive bias rather than a constraint imposed by a continuous penalty. This directly supports our central “unsolicited advice” hypothesis: when grounding is harmonized with the main objective, the model retains and exploits it naturally, rather than overwriting it once supervision is removed.
> > > > >
> > > > > We hope this clarifies our position within the literature and highlights the core contributions of the work. If any technical concerns remain that prevent a higher confidence score, we would be very grateful for further feedback and the opportunity to address them.
> > > > >
> > > > > | Curriculum Scheduling | RefCOCO (val) | RefCOCO+ (val) | RefCOCOg (val) |
> > > > > |---|----|----|---|
> > > > > | ❌ | 70.3 | 61.5 | 65.7 |
> > > > > | ✅ | 72.9 | 64.9 | 68.9 |

---

> ### Author Response · Authors · 2025-11-24
> **[Response 1] [Part 3] Lightweight and Scalable Data Curation Pipeline**
>
> **3) Data Curation Pipeline:**
>
> ---
>
> **3.1. Replacing GPT:**
>
> We thank the reviewer for raising this point. We agree that scalability and reproducibility are essential. In response to this concern, we have already removed all GPT usage from our pipeline.
> GPT was originally used in two places, but both have now been replaced with strong, fully open-source models:
>
> 1.	**Referring phrase generation.**
> This step is straightforward (extracting the visual entity and generating a short grounding phrase from a Q&A pair).
> We initially used GPT-4o-mini, but recent open-source models such as Qwen3-VL and InternVL3.5 achieve comparable GPT-5–level performance on many benchmarks. We therefore replaced this step entirely with InternVL3.5, making the pipeline fully reproducible.
>
> 2.	**Judge model for non-consensus cases.**
> The second use of GPT was in the mask-evaluation step, where we need the model to decide which of the four candidate masks best matches the phrase when IoU consensus fails.
> We also replaced GPT-4o with InternVL3.5 here. Although InternVL occasionally accepts imperfect masks in very challenging cases, the overall accuracy remains high.
>
> **Human Evaluation** To quantify the robustness of the updated pipeline, we conducted a 10K-sample human evaluation study. The results show that our updated pipeline achieves 93% accuracy, compared to 96% when using GPT. This indicates that the fully open-source version retains strong reliability while being substantially more practical and accessible.
>
> All updates have been incorporated into the paper text, Figure 4, and Table 3, and we provide more details in Appendix B.2. We also include Figure 6, which illustrates the GUI used during human evaluation.
>
> Overall, the pipeline is now lighter, fully open-source, reproducible, and scalable, addressing the reviewer’s concerns directly.
>
> ---
>
> **3.2. Complexity Analysis:**
>
> We appreciate the reviewer’s concern regarding the practicality of the data collection pipeline. We fully agree that a scalable and efficient procedure is crucial. Importantly, our pipeline is designed to be practical: using a single node (8×A100), we generated the full 2.5M masks in roughly one day, as the four grounding models can run in parallel and the judge model operates concurrently over only ~50% of the samples.
>
> Moreover, we run InternVL3.5 using the vLLM inference engine, which yields 6× faster throughput compared to standard HuggingFace Transformers, further reducing latency.
>
> It is also worth noting that our pipeline is significantly simpler and more efficient than the four-level GLaMM data-generation pipeline, which relies on a large collection of heterogeneous models—CO-DETR, EVA, OWL-ViT, POMP, GRiT, RAM Tag2Text, MDETR, BLIP-2, LLaVA, GPT4-RoI, and others—and requires constructing multi-stage scene graphs, dense captions, and hierarchical context.
>
> This design was appropriate at its time, but it is now outdated relative to the rapid progress in multimodal modeling. Modern VLMs such as Qwen3-VL, and InternVL3.5 are far stronger, enabling us to replace the entire multi-level pipeline with a much leaner, two-step procedure (grounding + lightweight judging) while achieving high quality.
>
> Our human study confirms that the resulting pseudo-masks are 93% accurate, nearly matching the GPT-based variant (96%) but with a dramatically simpler and faster pipeline.
>
> Overall, the updated pipeline is open-source, scalable, easy to reproduce, and substantially more efficient than prior large-scale grounding pipelines.
>
> | Component / Model         | Task                                   | Throughput                | Estimated Time for 2.5M Samples (1*A100)    | Estimated Time for 2.5M Samples (8*A100)                                                          |
> |-----|------|------|---------|---------------------|
> | **Sa2VA**                 | Segmentation grounding                  | ~5.6 samples/sec          | ≈ 125 hours (≈ 5.2 days)             | ≈ 16 hours  |
> | **GLaMM**                 | Segmentation grounding                  | ~22 samples/sec           | ≈ 31 hours (≈ 1.3 days)              | ≈ 4 hours  |
> | **OWL-V2**                | Open-vocabulary grounding               | ~66.6 samples/sec         | ≈ 10.4 hours                         | ≈ 1.3 hours |
> | **Grounding DINO**        | Open-vocabulary grounding               | ~66.6 samples/sec         | ≈ 10.4 hours      | ≈ 1.3 hours |
> | **InternVL3.5 (Judge)**   | Validates 50% of samples (non-consensus)| ~13.9 samples/sec         | ≈ 25 hours (≈ 1 day) | ≈ 3 hours |
> | **InternVL3.5 (Phrase)**  | Generates referring phrases             | ~41.7 samples/sec         | ≈ 16.7 hours                         | ≈ 2 hours |

---

### Comment · Area_Chair_MeNi · 2025-11-23
**Reviewer & Author Discussion**

Hi Reviewers,

Please kinly and actively participate in the review-author dicussion, raise your further concerns so that the authors can explain more, and make your final decisions.

---

### Meta-Review · Area_Chair_y4a8 · 2025-12-28

**Summary:**

This submission received diverse scores (6642). The majors concerns raised by reviewers are about novelty of the approach, strength of empirical validation, and plausibility of the data collection pipeline. After discussion, responses to major concerns from one of the reviewer were not responded. Even one reviewer would have changed her/his score, the averaged score remains under the bar. As the authors investigated an important and interesting question, I suggest them to carefully consider the concerns raised by reviewers and submit their work to the next conference.

**Reviewer Concerns:**

Some of the reviewer concerns were addressed by the rebuttal, but others were not confirmed by the reviewers.

**Reviewer Scores:**

If they had been able to participate fully in the discussion, one reviewer would have changed their score but there is no evidence to show all reviewers assigned negative scores would have changed their scores.

---

### Decision · Program_Chairs · 2026-01-26

Reject